# Targeted molecular profiling of rare olfactory sensory neurons identifies fate, wiring, and functional determinants

J Roman Arguello[1,2,3‡], Liliane Abuin[1‡], Jan Armida[1†], Kaan Mika[1†], Phing Chian Chai[1], Richard Benton[1*]

[1]Center for Integrative Genomics Faculty of Biology and Medicine University of Lausanne, Lausanne, Switzerland; [2]Department of Ecology and Evolution Faculty of Biology and Medicine University of Lausanne, Lausanne, Switzerland; [3]Swiss Institute of Bioinformatics, Lausanne, Switzerland

**Abstract** Determining the molecular properties of neurons is essential to understand their development, function and evolution. Using Targeted DamID (TaDa), we characterize RNA polymerase II occupancy and chromatin accessibility in selected *Ionotropic receptor* (*Ir*)-expressing olfactory sensory neurons in *Drosophila*. Although individual populations represent a minute fraction of cells, TaDa is sufficiently sensitive and specific to identify the expected receptor genes. Unique *Ir* expression is not consistently associated with differences in chromatin accessibility, but rather to distinct transcription factor profiles. Genes that are heterogeneously expressed across populations are enriched for neurodevelopmental factors, and we identify functions for the POU-domain protein Pdm3 as a genetic switch of Ir neuron fate, and the atypical cadherin Flamingo in segregation of neurons into discrete glomeruli. Together this study reveals the effectiveness of TaDa in profiling rare neural populations, identifies new roles for a transcription factor and a neuronal guidance molecule, and provides valuable datasets for future exploration.

*For correspondence:
Richard.Benton@unil.ch

[†]These authors also contributed equally to this work
[‡]These authors also contributed equally to this work

Competing interests: The authors declare that no competing interests exist.

## Introduction

Nervous systems are composed of vast numbers of neuron classes with specific structural and functional characteristics. Determining the molecular basis of these properties is essential to understand their emergence during development and contribution to neural circuit activity, as well as to appreciate how neurons exhibit plasticity over both short and evolutionary timescales.

The olfactory system of adult *Drosophila melanogaster* is a powerful model to identify and link molecular features of neurons to their anatomy and physiology. The *Drosophila* nose (antenna) contains ~50 classes of olfactory sensory neurons (OSNs), which develop from sensory organ precursor cells specified in the larval antennal imaginal disc (*Barish and Volkan, 2015*; *Jefferis and Hummel, 2006*; *Yan et al., 2020*). Each OSN is characterized by the expression of a unique olfactory receptor (or, occasionally, receptors), of the Odorant Receptor (OR) or Ionotropic Receptor (IR) families. These proteins function – together with broadly expressed, family-specific co-receptors – to define neuronal odor-response properties (*Benton, 2015*; *Gomez-Diaz et al., 2018*; *Robertson, 2019*; *Rytz et al., 2013*). Moreover, the axons of all neurons expressing the same receptor(s) converge to a discrete and stereotyped glomerulus within the primary olfactory center (antennal lobe) in the brain, where they synapse with both local interneurons (LNs) and projection neurons (PNs) (*Brochtrup and Hummel, 2011*; *Hong and Luo, 2014*; *Jefferis and Hummel, 2006*). While the unique properties of individual OSN classes are assumed to reflect distinct patterns of gene expression, it is unclear how many molecular differences exist between sensory neuron populations,

which types of genes (beyond receptors) distinguish OSN classes, and whether gene expression differences reflect population-specific chromatin architecture.

Exploration of the molecular properties of specific neuron types has been revolutionized by single-cell RNA sequencing (scRNA-seq) technologies (*Svensson et al., 2018*). However, the peripheral olfactory system of *Drosophila* poses several challenges to cell isolation: OSNs are very small (~2–3 μm soma diameter) and up to four neurons are tightly embedded under sensory sensilla (a porous cuticular hair that houses the neurons' dendrites) along with several support cells that seal each neuronal cluster from neighboring sensilla (*Shanbhag et al., 1999*; *Shanbhag et al., 2000*). Furthermore, each OSN class represents only a very small fraction of cells of the whole antenna, as few as 5–10 neurons of a total of ~3000 cells (both neuronal and non-neuronal) (*Grabe et al., 2016*; *Shanbhag et al., 1999*). Recent work (*Li et al., 2020*) – discussed below – successfully profiled RNA levels in OSNs at a mid-developmental time-point (42–48 hr after puparium formation [APF]), before these cells become encased in the mature cuticle. In addition, a preprint describing successful transcriptomic profiling of single nuclei of adult OSNs (*McLaughlin et al., 2020*) will also be considered later.

In this work, we used the Targeted-DNA adenine methyltransferase identification (TaDa) method (*Southall et al., 2013*) to characterize the molecular profile of specific populations of OSNs. TaDa entails expression of an RNA polymerase II subunit (RpII15, hereafter PolII) fused to the *E. coli* DNA adenine methyltransferase (Dam). This complex (or, as control, Dam alone) is expressed within a specific cell population ('targeted') under the regulation of a cell type-specific promoter using the Gal4/UAS system. When bound to DNA, Dam:PolII methylates adenine within GATC motifs; enrichment of methylation at genes compared to free Dam control samples provides a measure of PolII occupancy, which is correlated with genes' transcriptional activity (*Southall et al., 2013*; *Figure 1A*). In addition to indicating PolII occupancy within a given cell type, TaDa datasets can also provide information on chromatin accessibility (CATaDa) (*Aughey et al., 2018*; *Figure 1A*). Analysis of the methylation patterns across the genome generated by untethered Dam is assumed to reflect the openness of chromatin, analogous to – and in good agreement with – ATAC-seq datasets (*Aughey et al., 2018*; *Buenrostro et al., 2013*).

The selectivity of expression of Dam:PolII or Dam avoids the requirement for cell isolation prior to genomic DNA extraction. Previous applications of TaDa in *Drosophila* have profiled relatively abundant cell populations in the central brain, intestine and male germline (*Doupé et al., 2018*; *Southall et al., 2013*; *Tamirisa et al., 2018*; *Widmer et al., 2018*). Although OSN classes can be effectively labeled using transgenes containing promoters for the corresponding receptor genes, it was unclear whether TaDa would be sufficiently sensitive to measure transcriptional activity from an individual population of sensory neurons within the antenna. We therefore set out to profile several classes of IR-expressing OSNs to test the approach and identify novel factors that distinguish the properties of these evolutionarily closely related neuronal subtypes.

## Results

### Targeted DamID of OSN populations

To test the feasibility of TaDa for profiling *Drosophila* OSNs, we first focused on the Ir64a population, which comprises ~16 neurons that are housed in sensilla in the sacculus (a chamber within the antenna), where they detect acidic odors (*Ai et al., 2010*). We first verified that *Ir64a promoter-Gal4* (hereafter, *Ir64a-Gal4*) induction of Dam or Dam:PolII did not impact the expression or localization of the IR64a receptor (*Figure 1B*) or the projection of these neurons to the antennal lobe (*Figure 1C*). We proceeded to collect triplicate samples of ~2000 adult antennae and processed these for TaDa (see Materials and methods). This experimental design is expected to integrate PolII occupancy patterns from the mid-pupal stage (when *Ir*-Gal4 expression initiates) to adults. We therefore anticipated identifying genes that are differentially expressed both during the second half of OSN development and in mature antennae. As a positive control, we first examined the mapping of reads of methylated DNA fragments to the *Ir64a* gene. Dam:PolII consistently accessed this gene region at higher levels than Dam alone, particularly near the 5' end of the locus (*Figure 1D*), consistent with previously described genome-wide PolII occupancy patterns (*Southall et al., 2013*).

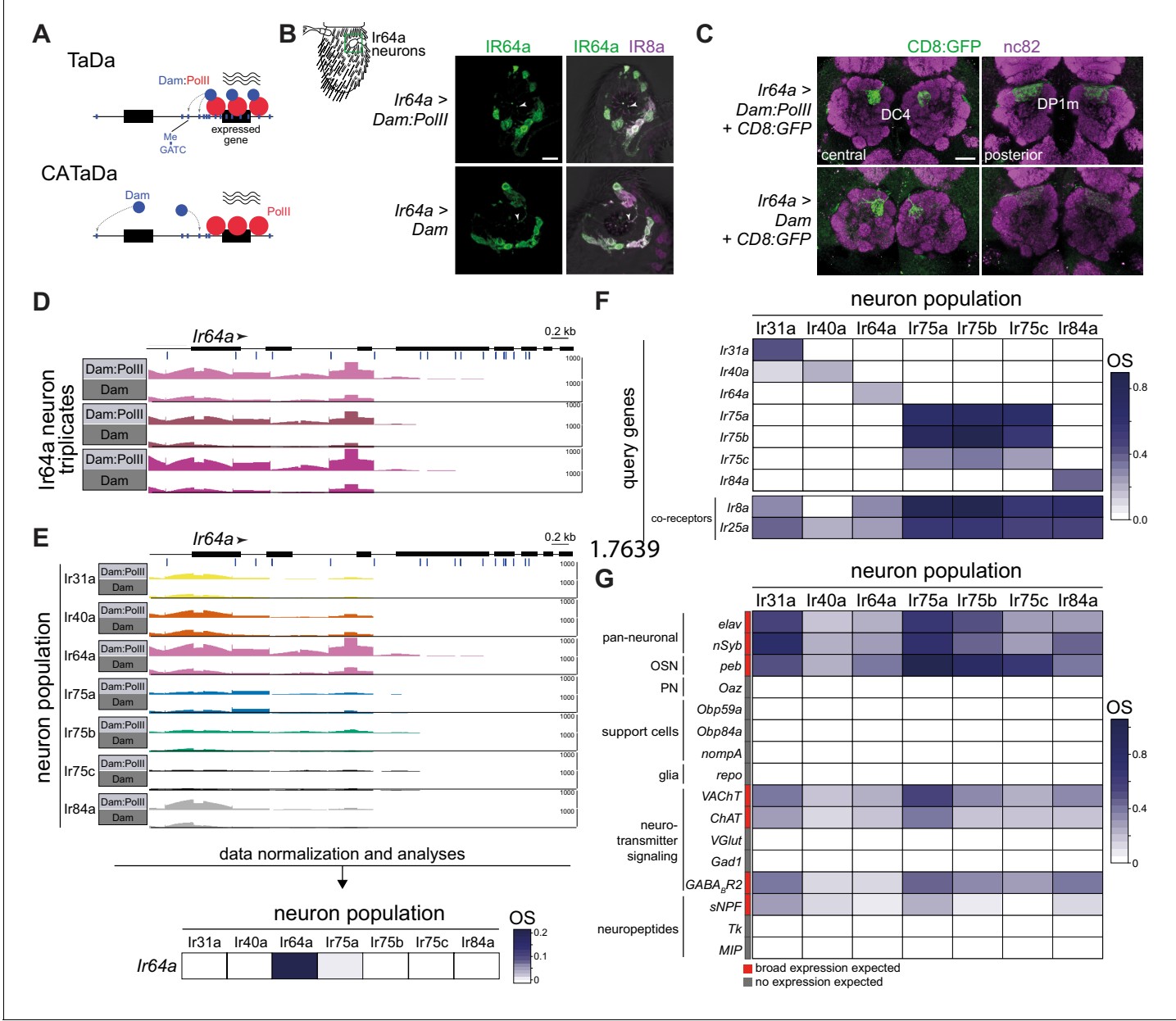

**Figure 1.** Targeted DamID of OSN populations. (**A**) Schematic of the principles of Targeted DamID (TaDa) and Chromatin accessibility TaDa (CATaDa). TaDa reports on the enrichment of GATC methylation (Me; blue lines) by a Dam:PolII fusion (top) relative to Dam alone (bottom), thereby identifying PolII-transcribed genes. In CATaDa, analysis of methylation patterns of free Dam provides information on chromatin accessibility. (**B**) Immunofluorescence for IR64a and IR8a on antennal sections of animals expressing in Ir64a neurons either Dam:PolII (*Ir64a-Gal4/+;UAS-LT3-Dam: RpII15/+*) or Dam alone (*Ir64a-Gal4/+;UAS-LT3-Dam/+*). The approximate field of view shown is indicated by the green box on the antennal schematic on the left. The merged fluorescent channels are overlaid on a bright-field background to reveal anatomical landmarks. The arrowheads point to IR64a that is localized in the neuron sensory endings within the sensillar hairs. Scale bar = 10 μm. (**C**) Immunofluorescence for GFP and the neuropil marker nc82 on whole-mount brains of animals expressing in Ir64a neurons a CD8:GFP reporter together with Dam:PolII (*Ir64a-Gal4/+;UAS-LT3-Dam:RpII15/ UAS-mCD8:GFP*) or free Dam (*Ir64a-Gal4/+;UAS-LT3-Dam/UAS-mCD8:GFP*). Two focal planes are shown to reveal the two glomeruli (DC4 and DP1m) innervated by Ir64a neurons. Background fluorescence in the GFP channel was slightly higher across the brain in the Dam-alone samples. Scale bar = 20 μm. (**D**) Illustration of raw sequence read data underlying the TaDa analyses. Bar heights indicate the read coverage of methylated GATC fragments mapping to the *Ir64a* gene (transcribed left-to-right, as indicated by the arrowhead; blue lines under the gene model indicate the position of GATC motifs). The precise transcription start site of *Ir64a* is unknown. For each of three replicate experiments, the top row is the Dam:PolII sample, which has higher read counts compared to the Dam-alone sample below. The y-axes (range 0–1000) indicate the read depth. (**E**) Top: comparisons among the seven Ir neuron datasets of the raw sequence reads mapping to the *Ir64a* gene for the Dam:PolII and Dam samples (a single representative replicate is shown for each). Genotypes are of the form shown in (**B**), except for Ir75a and Ir75c neuron populations where third chromosome *Ir-Gal4* driver

*Figure 1 continued on next page*

*Figure 1 continued*

transgenes were used. Only the Ir64a population displays a higher read count for Dam:PolII samples compared to Dam-alone samples. The y-axes (range 0–1000) indicates the read depth. Bottom: heatmap representation of the occupancy scores (OSs) following data normalization and statistical analyses (based on data across the full gene body (see Materials and methods); color scale on the right) for *Ir64a* calculated across triplicates for each of the seven neuron populations. (F) OS heatmap for the seven target *Ir* genes, as well as the main *Ir* co-receptor genes, in the seven Ir neuron populations. OSs and FDR values associated with each *Ir* gene within the corresponding neuron population are shown in **Supplementary file 5**. (G) OS heatmap of diverse positive- and negative-control genes in the seven Ir neuron populations (i.e. with known or expected expression/lack of expression). Abbreviations are defined in the main text. See also **Supplementary file 1** and DamID_files.zip (https://gitlab.com/roman.arguello/ir_tada/-/tree/main/DamID_analyses) for data.

The online version of this article includes the following figure supplement(s) for figure 1:

**Figure supplement 1.** *Ir* gene Occupancy Scores across Ir populations.
**Figure supplement 2.** *Or* gene Occupancy Scores across Ir populations.
**Figure supplement 3.** *Gr* gene Occupancy Scores across Ir populations.
**Figure supplement 4.** *Obp* gene Occupancy Scores across Ir populations.

We extended the TaDa approach using drivers to target six additional individual OSN classes, including all acid-sensing Ir populations (Ir31a, Ir75a, Ir75b, Ir75c, Ir84a) (**Grosjean et al., 2011**; **Silbering et al., 2011**), as well as the hygrosensory Ir40a neurons (**Enjin et al., 2016**; **Knecht et al., 2016**); these populations range from ~8 to ~25 cells per antenna (**Knecht et al., 2016**; **Prieto-Godino et al., 2017**). Reproducibility among all triplicate datasets (average $r^2$ = 0.85) was consistent with previous TaDa studies (**Southall et al., 2013**). As expected, the enrichment of Dam:PolII access to the *Ir64a* gene was specific to the Ir64a neurons (**Figure 1E**). We quantified 'PolII occupancy scores' (OSs) by calculating the $\log_2$ ratio of the normalized number of reads from the Dam:PolII samples compared to the normalized number of reads in control Dam samples across the entire gene body. In Ir64a neurons, the *Ir64a* gene OS was significantly positive (OS = 0.20; False Discovery Rate (FDR) = 4.26e-10), while it was not significantly different from zero in the other six neuron populations (**Figure 1E**, bottom).

Similarly, OSs for other *Ir* genes were significantly positive in the corresponding neuronal population, but not in those in which the receptors are not expressed (**Figure 1F**). We noted two exceptions: first, in Ir31a neurons, *Ir40a* had a positive OS (0.12; FDR = 1.93e-13), although it was much lower than that of *Ir31a* (OS = 0.41; FDR = 2.67e-6) (**Figure 1F**). Second, the *Ir75a*, *Ir75b*, and *Ir75c* genes had positive OSs in all three of the corresponding neuron populations (**Figure 1F**). As the proteins they encode (and the corresponding *Ir-Gal4* drivers) are expressed in discrete neuron populations, this latter result is most likely due to these tandemly clustered genes having overlapping transcription units (**Prieto-Godino et al., 2017**).

To further examine the specificity of TaDa in reporting on neuron population-specific gene expression, we plotted OSs for all members of the *Ir* repertoire, as well as *Or* and *Gustatory receptor* (*Gr*) families. The vast majority of these genes are expressed only in other antennal sensory neuron populations, distinct chemosensory organs and/or life stages (**Chen and Dahanukar, 2020**; **Joseph and Carlson, 2015**; **Sánchez-Alcañiz et al., 2018**). Concordantly, only very few genes show significant OSs in any of the seven populations of Ir neurons analyzed by TaDa (**Figure 1F**, **Figure 1— figure supplements 1–3**). Two prominent examples are *Ir8a* and *Ir25a*, which encode co-receptors for subsets of tuning IRs (**Abuin et al., 2011**). *Ir8a* has a significant OS in all Ir populations except for Ir40a neurons, which is consistent with the selective expression and function of IR8a in acid-sensing OSNs but not hygrosensory neurons (**Abuin et al., 2011**; **Figure 1F**). IR25a is broadly expressed in most antennal neurons (**Abuin et al., 2011**; **Benton et al., 2009**) – although it may have a role in only a subset of these – which is reflected in a significant OS for *Ir25a* in all seven populations (**Figure 1F**). IR40a is co-expressed with another receptor, IR93a (**Enjin et al., 2016**; **Knecht et al., 2016**), although Ir40a neurons display significant occupancy of *Ir93a*, several other Ir populations also have, unexpectedly, a significant OS for this gene (**Figure 1—figure supplement 1**). TaDa signals of real transcription of *Ir93a* may be confounded by an intronic *Doc* transposable element.

Of the other chemosensory receptor genes displaying unexpected significant OSs in these Ir neurons (**Figure 1—figure supplements 1–3**), many are located in the introns of other genes (e.g. *Ir47a*, within the *slowpoke 2* locus, which encodes a nervous-system expressed potassium channel [**Budelli et al., 2016**]), or have overlapping transcripts with other genes (e.g. *Gr2a*, which overlaps

with *futsch*, a gene encoding a neuronal microtubule-binding protein [*Hummel et al., 2000*]). In these cases, calculation of a specific OS for the chemosensory genes is impossible, and we suspect that the OS reflects transcription of the non-receptor gene. A similar issue might affect receptor genes that are in very close proximity to (though not overlapping with) other genes, given that TaDa signals depend upon GATC genomic distributions and not transcript boundaries.

Olfactory receptors have lower transcript levels in the antenna compared to other classes of genes with broader neuronal or non-neuronal expression patterns (*Menuz et al., 2014*; *Shiao et al., 2015*). We extended the examination of the TaDa datasets to other genes that are known to be expressed (or not expressed) in the analyzed Ir neuron populations (*Figure 1G*). Two broadly-expressed neuronal genes, *elav* and *nSyb*, display significant OSs across populations, as does *pebbled* (*peb*), a molecular marker for OSNs (*Sweeney et al., 2007*). By contrast, *oaz*, a broad marker of PNs (*Hong et al., 2009*), is not significantly occupied. Many of the most highly expressed genes in the antenna are those encoding Odorant Binding Proteins (OBPs), which are thought to be transcribed exclusively in non-neuronal cells (*Larter et al., 2016*), often at >10 fold higher levels than olfactory receptor genes, as determined by bulk RNA-seq (*Menuz et al., 2014*; *Shiao et al., 2015*). (This idea was challenged recently by the OSN snRNA-seq analysis [*McLaughlin et al., 2020*], which will be considered further in the Discussion). Two of these, *Obp59a* and *Obp84a* are expressed in sensilla housing Ir neurons (*Larter et al., 2016*; *Sun et al., 2018a*), but neither of these genes displays significant OSs in any neuron population (*Figure 1G*). Such absence of occupancy is true for the *Obp* repertoire as a whole (*Figure 1—figure supplement 4*). Similarly, markers for other support cell types, such as *nompA* (thecogen cell) (*Chung et al., 2001*; *Larter et al., 2016*), and the glial marker *repo* are not significantly occupied (*Figure 1G*).

Electrophysiological experiments indicate that antennal OSNs are cholinergic (*Kazama and Wilson, 2008*). In line with this functional property, genes encoding the vesicular acetylcholine transporter (*VAChT*) and choline acetyltransferase (*ChAT*) have significant OSs in all populations, while glutamatergic and GABAergic neuron markers (vesicular glutamate transporter (*VGlut*) and glutamic acid decarboxylase 1 (*Gad1*)) do not (*Figure 1G*). We note, however, that a GABA receptor subunit (encoded by *GABA-B-R2*) – which is heterogeneously expressed and mediates population-specific presynaptic gain control in different Or OSNs (*Root et al., 2008*) – displays significant, but variable, OSs in different Ir populations (*Figure 1G*). Finally, of the multiple neuropeptides detected in the antennal lobe by proteomics, only one, short neuropeptide F (sNPF), originates from OSNs (*Carlsson et al., 2010*), where it has a role in glomerular-specific presynaptic facilitation (*Root et al., 2011*). Consistently, we observe significant (and heterogeneous) OSs for *sNPF*, but not for genes encoding neuropeptides whose presence in the antennal lobe derives from expression in central neurons (e.g. Tachykinin [Tk], Myoinhibiting peptide [Mip]) (*Carlsson et al., 2010*).

Together, these results indicate that TaDa is sufficiently sensitive and specific to detect PolII occupancy at transcriptionally active genes in very small populations of neurons embedded within the cellularly-complex antennal tissue.

## Global analysis of TaDa datasets

We next surveyed OSs at a genome-wide level, to characterize patterns of transcriptional activity in different Ir neuron populations. The number of genes significantly occupied by PolII ranged from 3603 to 4775 across the seven datasets, representing 20–27% of all genes (*Figure 2A* and *Supplementary files 1–3*). To investigate patterns of similarity in global OSs among these populations, we carried out hierarchical clustering of the mean OSs for all genes in the genome (*Figure 2B*). This analysis revealed an interesting correspondence in the clustering of OSs with the phylogeny of the corresponding receptor genes (*Prieto-Godino et al., 2017*): neurons expressing the most recently duplicated receptor genes, *Ir75b* and *Ir75c*, were clustered together. The next most similar neuron population expresses *Ir75a*, which is the likely ancestral member of this tandem gene array (*Prieto-Godino et al., 2017*). Ir40a and Ir64a neurons clustered together away from the other acid-sensing populations, possibly reflecting the development of these neurons within the sacculus, rather than in sensilla located on the antennal surface.

Beyond the unique patterns of sensory receptor expression, little is known about the nature or neural specificity of other signaling pathways in OSNs. To gain insight into these properties, we examined the occupancy of genes encoding putative neuropeptides, G protein-coupled receptors (GPCRs) and ion channels (excluding chemosensory receptors) in the seven Ir populations

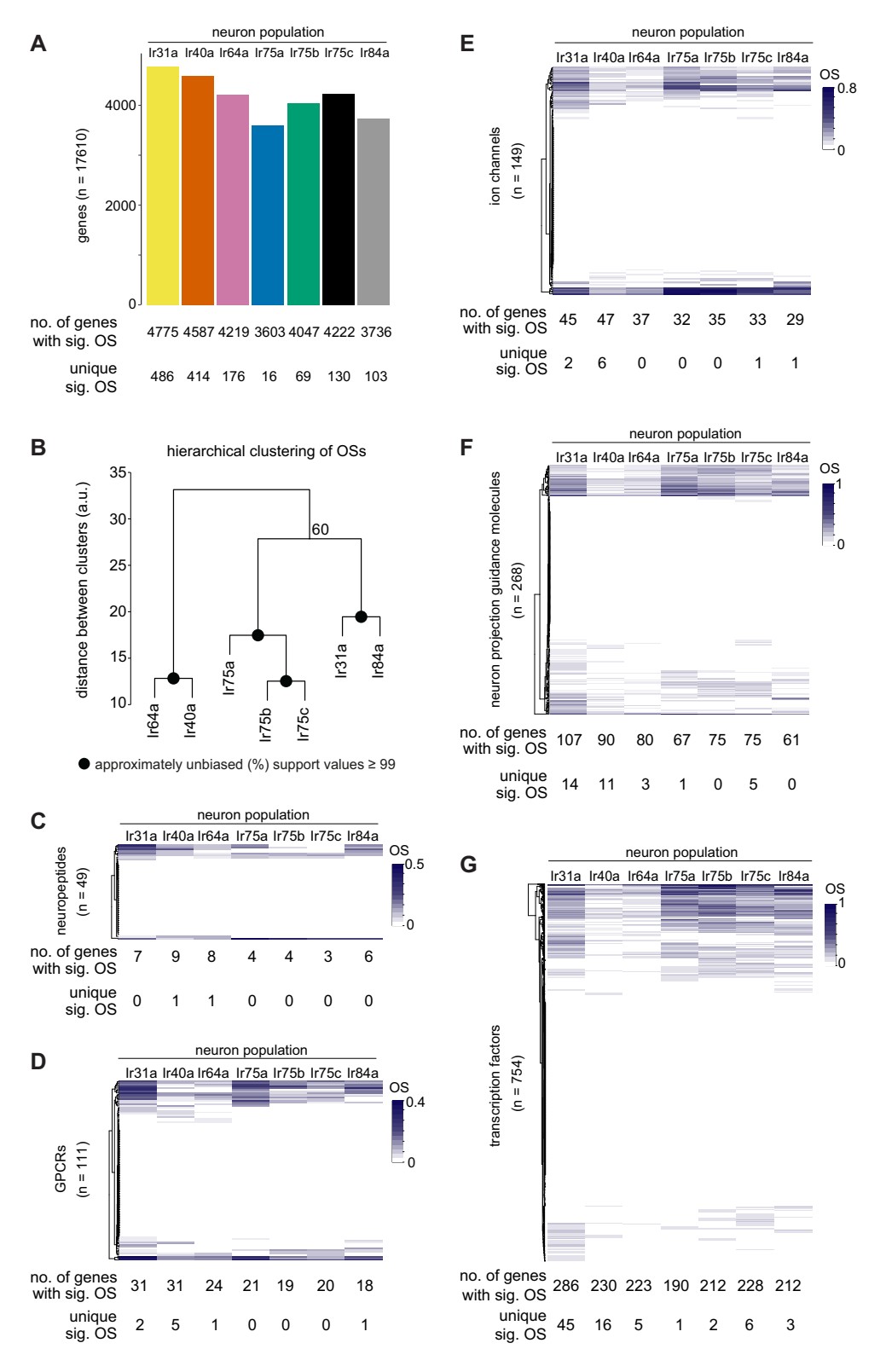

**Figure 2.** Global analyses of TaDa datasets. (**A**) Bar plot summarizing the number of genes in the genome that have significantly positive OSs within each Ir neuron population, and the number of significantly occupied genes that are unique to a given neuron population. See also *Supplementary files 1–3*. (**B**) Hierarchical clustering (arbitrary units, a.u.) of the seven Ir neuron populations based upon the OSs of all genes. All nodes had approximately unbiased support values of 100% except for the one denoted with 60%. (**C–G**) Heatmaps displaying the hierarchical clustering of OSs for the genes

*Figure 2 continued on next page*

*Figure 2 continued*

encoding (C) neuropeptides, (D) GPCRs, (E) ion channels, (F) neuron projection guidance molecules, and (G) transcription factors. For each population, the total number of genes with a significant OS, and the number of those that are unique to a given population, are shown below (see also *Supplementary files 6–20*) which list, for each category: (i) all analyzed genes, (ii) significantly occupied genes, and (iii) population-specific occupied genes.

The online version of this article includes the following figure supplement(s) for figure 2:

**Figure supplement 1.** Heterogeneous expression of Engrailed in Ir neuron populations.

(*Figure 2C–E* and *Supplementary files 6–14*). We observed that each Ir neuron population displays PolII occupancy of a distinct combination of 3–9 neuropeptide genes (out of 49 in the genome), 18–31 GPCR genes (out of 111), and 29–47 ion channel genes (out of 149). There were, however, only a few cases of genes that are unique to a single neuronal population (*Supplementary files 8*, *11* and *14*).

OSNs are also distinguished by their morphological properties, notably their projections to discrete glomeruli within the antennal lobe. Genetic studies have identified several secreted and transmembrane proteins that play roles in different steps of the neuronal guidance process (*Brochtrup and Hummel, 2011*; *Hong and Luo, 2014*; *Jefferis and Hummel, 2006*), but our understanding remains incomplete and many key molecules likely remain to be discovered. Although our TaDa experiments profile the latter stages of OSN development – after glomerular targeting has occurred (*Jefferis and Hummel, 2006*) – the expression of known OSN axon guidance genes persists into adult stages (*Barish et al., 2018*; *Menuz et al., 2014*). We therefore reasoned our datasets could permit surveying of the potential complexity and neuron-type specificity of the molecular machinery underlying this process. We examined OSs of ~268 genes encoding molecules with known or implicated roles in neuron projection guidance (*Figure 2F* and *Supplementary files 15–17*). While each neuron population had a unique combination of many dozens of PolII-occupied genes, individual datasets had very few or no unique genes (*Figure 2F* and *Supplementary file 17*). These observations suggest that different OSN populations depend upon specific combinations of largelyoverlapping sets of guidance factors.

The expression of both chemosensory receptors and guidance molecules depends upon regulatory networks of transcription factors (*Barish and Volkan, 2015*; *Jafari et al., 2012*). We therefore assessed global patterns of PolII occupancy of the set of 754 predicted *Drosophila* transcription factors (*Pfreundt et al., 2010*) across the seven Ir populations. Half of these genes (379/754) displayed significant occupancy in at least one population of Ir neurons, with individual OSN classes having 190–286 occupied transcription factor genes (*Figure 2G* and *Supplementary files 18–20*). There was considerable variation in Dam:PolII occupancy of transcription factor genes across the neuron populations (*Figure 2G*), consistent with each neuronal subtype possessing a unique and complex gene regulatory network to support its specific differentiation properties. Here, Ir31a neurons stood out (45 uniquely occupied genes); the reasons for this are unclear, as the biology of this sensory pathway is poorly understood. By contrast, Ir75a neurons had a single uniquely occupied transcription factor gene (*pou domain motif 3* (*pdm3*), investigated further below).

Many of the transcription factors with known functions in OSN development are broadly-expressed across OSNs, such as Acj6, Fer1, and Onecut (*Clyne et al., 1999*; *Jafari et al., 2012*); consistently, these genes display positive OSs in all seven neuron populations (*Supplementary file 19*). One exception is Engrailed (En), which is expressed and required in a subset of OSNs (*Chou et al., 2010b*; *Song et al., 2012*). In our TaDa datasets, we observed that *en* displays significant OSs in Ir31a, Ir40a, and Ir64a populations (*Figure 2—figure supplement 1A*), concordant with the robust detection of En protein in only these three Ir neuron classes (*Figure 2—figure supplement 1B–C*).

## Comparative chromatin accessibility in OSNs

We next performed CATaDa analysis through investigation of the Dam-alone datasets, by using these to generate chromatin accessibility maps similar to ATAC-seq and FAIRE-seq (*Aughey et al., 2018*). Genome-wide visualization of the peaks of methylated sequences across the genome (see Materials and methods) revealed heterogeneous patterns of chromatin accessibility, with the

expected overall decrease in open chromatin toward the heterochromatic centromeric regions of

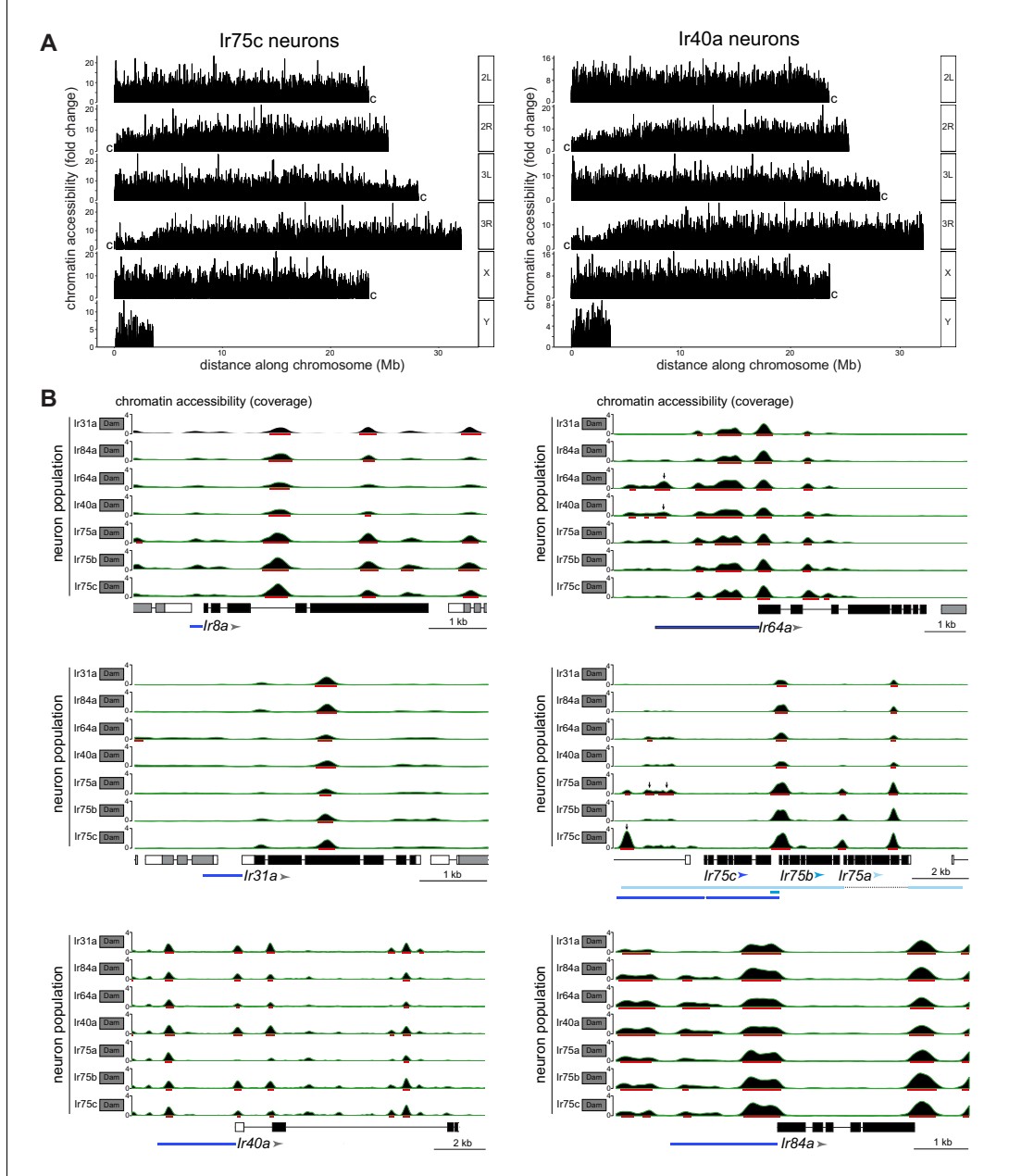

**Figure 3.** CATaDa reveals global similarity in chromatin accessibility at *Ir* genes across Ir neuron populations. (**A**) Comparison of chromatin accessibility peak locations (q < 0.05) over the *D. melanogaster* genome for two example neuron populations, Ir75c and Ir40a. The y-axis represents the fold enrichment for the peak summit against random Poisson distribution of the small local region (1000 bp). 2L/3L and 2R/3R refer to left and right arms of chromosomes 2/3; 'c' at the end of each arm indicates the centromeric (heterochromatic) end of the arm (the Y chromosome is mainly heterochromatic). (**B**) Plots of *Ir* gene exons are shown in black, and UTRs in white; exons of flanking genes are shown in gray. The arrowheads indicate the direction of *Ir* transcription. The minimal defined regulatory sequences to recapitulate *Ir* expression are indicated with blue bars (color-coordinated with the arrowheads for *Ir75a*, *Ir75b*, and *Ir75c*) (**Prieto-Godino et al., 2017**; **Silbering et al., 2011**). The y-axis scale shows the CATaDa accessibility at the genomic region with the RPGC normalization method. The red bars are the significant peaks with q < 0.05. In the *Ir75a/Ir75b/Ir75c* genomic region, the arrows mark significant peaks that are unique to the corresponding Ir neuron populations and which are contained within known regulatory regions of these *Ir* genes. In the *Ir64a* genomic region, the arrows mark significant peaks observed in both Ir64a and Ir40a neurons. Note that the smooth boundaries around peaks of Dam-accessible regions are visual artifacts of the analysis; they do not reflect the resolution of CATaDa, which is discrete at the GATC motifs.

the chromosome arms (*Figure 3A*).

To examine chromatin accessibility patterns at a gene level, we focused on CATaDa signals at the *Ir* loci that distinguish different populations – that is, the seven tuning *Ir* genes and *Ir8a* – whose cell type-specific transcriptional activity is well-established (*Abuin et al., 2011*; *Benton et al., 2009*; *Prieto-Godino et al., 2017*). For each gene, peaks were present at the presumed promoter region, as well as in upstream regions, which may represent enhancers. Comparisons across neuron populations revealed that peak distribution was similar across all eight genes (*Figure 3B*). This observation indicates that neuron-specific differences in *Ir* gene transcription are not due to overt differences in promoter and enhancer accessibility. However, we noted the presence of peaks upstream of *Ir75a* and *Ir75c* that are unique to their respective neuron populations (q-value <0.05). These peaks lie within regions corresponding to those that, when cloned into reporter transgenes, are sufficient to recapitulate receptor expression patterns (*Ai et al., 2010*; *Prieto-Godino et al., 2017*; *Silbering et al., 2011*; *Figure 3B*). Such peaks may therefore reflect enhancers that are accessible/ functional only in these Ir neuron populations. Peaks were also identified upstream of *Ir64a* in both Ir64a neurons and Ir40a neurons (*Figure 3B*). This observation suggests that these sacculus neuron populations share chromatin accessibility in this region but that other factors (e.g. *trans*-acting proteins) are required to determine the cell type-specific transcription of *Ir64a*.

## Identifying genes displaying differential expression in Ir populations

The analyses of genes encoding chemosensory receptors, transcription factors and other categories of proteins demonstrate the use of TaDa OSs to identify potential developmental and functional determinants that are expressed within specific OSN populations. While differences in OS for a given gene across populations may suggest heterogeneous expression levels, we sought to test formally for PolII occupancy differences between populations. We implemented a different analysis method that uses variation in read-depth along the gene body in the triplicate paired Dam-alone/Dam:PolII TaDa datasets (*Figure 4A* and Materials and methods). We identified candidates that had at least two GATC motifs for which the excess of Dam:PolII read-depth compared to the Dam-alone read-depth significantly varied in the same direction between two or more neuron populations (see Materials and methods). With this criterion, 1694 genes exhibited significant variation across the seven populations, indicating that only a relatively small proportion of genes (~8%) differ among these neuron types. This number is likely to be an overestimate as it includes overlapping and/or intronic genes that, as described above, cannot be parsed further using TaDa data. Importantly, this subset encompasses all of the expected *Ir* genes (see *Figure 4D*). To examine how each individual Ir neuron dataset varied in comparison to the six others, we also quantified significant differences between each pair of neuron populations (*Figure 4B*). Despite the methodological differences with the TaDa analysis that produces OSs, the proportion of genes contributing to the total number of pairwise difference approximates the clustering of global OSs (*Figure 2B*). For example, comparisons among the closely clustered Ir75a, Ir75b, and Ir75c neuron datasets result in very few of the total differences (0.4–1.7%). Conversely, slightly more than half (51%) of the total differences in the Ir84a neuron dataset arise from comparisons with the distantly clustered Ir40a neuron dataset.

We investigated the general characteristics of the 1694 differentially occupied genes by performing Gene Ontology analyses, finding highly significant enrichment in neuron, neural development, and signaling categories (*Figure 4C*). This enrichment is consistent with these genes underlying the known unique anatomical and physiological properties of these neuron classes, and indicates that our comparative TaDa approach would be valuable in identifying candidates that contribute functionally to these properties.

To establish a priority list for follow-up experiments, we ordered the 1694 genes by the total number of differentially occupied GATC motifs, reasoning that this would highlight those displaying differential occupancy along a broad region of the gene body (*Figure 4D*). While this 'ranking' is biased toward longer genes (which on average will have more GATC motifs), we emphasize that this does not exclude interest in genes with fewer GATC motifs; for example, *Ir* genes are distributed widely in this list (*Figure 4D*). We subsequently focused on experimental validation of the top-ranked transcription factor gene, *pdm3*, and the top-ranked gene encoding a neuron guidance molecule, *flamingo* (*fmi*; also known as *starry night* (*stan*), flybase.org/reports/FBgn0024836), as described below.

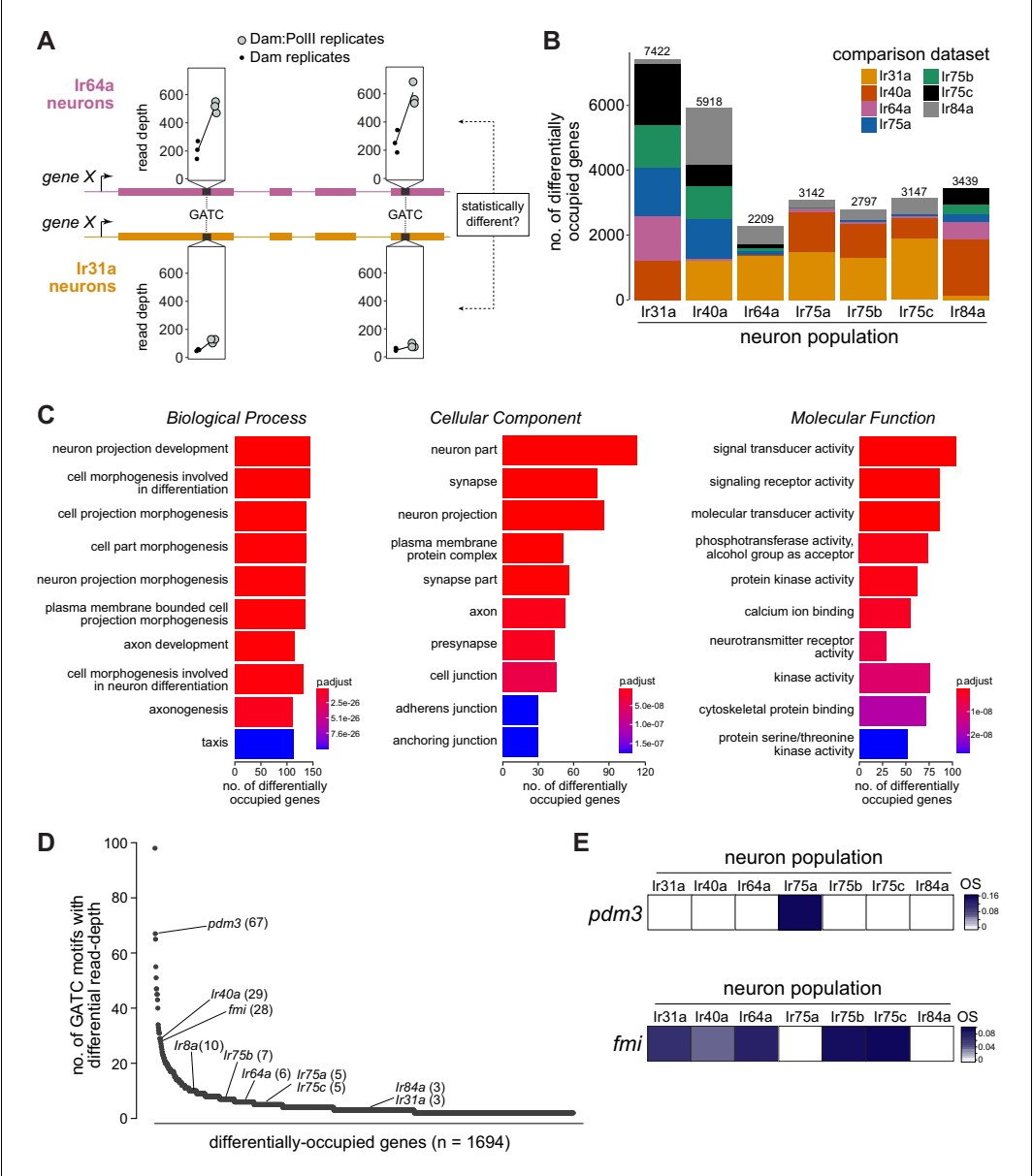

**Figure 4.** Differentially occupied genes across Ir populations. (**A**) Schematic illustrating the identification of a differentially occupied gene between two Ir populations by using variation in read depth at GATC motifs (see Materials and methods). In this example, occupancy of '*gene X*' is compared between Ir64a neurons and Ir31a neurons. At GATC motifs (black bars), a test is applied to determine if significant variation exists in the relative read depths (in triplicate Dam:PolII versus Dam-alone experiments) between these two neuron populations. Fictive data for two GATC motifs are depicted: for both, the relative read depth in the Dam:PolII experiments (compared to the Dam-alone experiments) is greater in the Ir64a neurons. (**B**) Stacked bar plots of the numbers of genes in the genome that are differentially occupied among Ir neuron populations based on pairwise tests (see Materials and methods). The colors indicate the proportions of genes emerging from comparisons with the six other neuron populations. The figures above the bars are the total numbers of genes from all pairwise comparisons; note that a given gene may be counted multiple times. (**C**) Gene Ontology (GO) analysis of the 1694 differentially occupied genes, showing the top ten over-represented GO terms. The x-axis represents the number of genes annotated to a particular GO term in the input subset. The value indicated by the colored bar is the probability of observing at least the same number of genes associated to that GO term compared to what would have occurred by chance. (**D**) Ranking of the 1694 differentially occupied genes ordered by the number of GATC motifs contributing to their between-neuron differences (uncorrected for gene length). (**E**) OS heatmaps (calculated as in *Figures 1–2*) of *pdm3* and *fmi* in the seven Ir neuron populations.

The online version of this article includes the following source data for figure 4:

**Source data 1.** Candidate genes for pairwise differential occupancy based on the Wald Test with DESeq2.
**Source data 2.** Candidate genes for differential occupancy based on the Likelihood Ratio Test within DESeq2.
**Source data 3.** Number of GATC motifs within candidate differentially expressed genes contributing to significant differences.

# Pdm3 acts as a genetic switch to distinguish Ir75a and Ir75b neuron fate

The robust signal displayed by *pdm3* for between-population differences is consistent with this gene having a significant OS only in Ir75a neurons (*Figure 4E*). Notably, from the global survey, *pdm3* is the sole transcription factor gene that is uniquely occupied in this neuron population (*Figure 2G* and *Supplementary file 20*). Pdm3 is a nervous-system enriched transcription factor (*Brown et al., 2014*; *Chen et al., 2012*; *Tichy et al., 2008*), although its precise in vivo binding specificity is unknown. In another olfactory organ, the maxillary palp, Pdm3 is expressed in multiple classes of Or neurons and functions in controlling receptor expression and axon guidance (*Tichy et al., 2008*), but its role in Ir neurons is unknown. Using Pdm3 antibodies, we first analyzed protein expression in antennae in which we co-labeled each of the Ir populations with the driver lines used in the TaDa experiments. Only Ir75a neurons displayed robust nuclear Pdm3 immunofluorescence, although a subset of Ir40a neurons had above-background signals (*Figure 5A–B*).

To examine the role of *pdm3* in Ir neurons we performed transgenic RNA interference (RNAi) using *peb-Gal4*. This driver is expressed broadly in OSNs from soon after their terminal divisions (~16 hr APF) throughout the initiation of olfactory receptor expression (~40–48 hr APF) (*Li et al., 2020*; *Sweeney et al., 2007*). *peb-Gal4*-driven pdm3$^{RNAi}$ (*peb>pdm3$^{RNAi}$*) led to loss of most IR75a expression, with the remaining detectable neurons expressing very low levels of this receptor (*Figure 5C–D*). By contrast, robust expression of all other Irs was maintained (*Figure 5C–D*). We noted, however, that the number of cells expressing IR75b increased in *pdm3$^{RNAi}$* antennae (*Figure 5D*). These observations suggest that Pdm3 functions (directly or indirectly) to promote IR75a expression and repress IR75b expression.

The novel IR75b-expressing cells in *pdm3$^{RNAi}$* antennae are mainly located in the proximal region of the antenna near the sacculus, where Ir75a neurons are normally found (*Figure 6A*). This distribution suggested that loss of Pdm3 results in a switch of receptor expression from IR75a to IR75b. We investigated this possibility by examining the activity of transcriptional reporters for these receptors, encompassing minimal regulatory DNA sequences fused to *CD4:tdGFP* (hereafter, *GFP*) (*Prieto-Godino et al., 2017*). Both *Ir75a-GFP* and *Ir75b-GFP* transgenes are faithfully expressed in IR75a and IR75b protein-expressing neurons (*Figure 6B–C*). In *pdm3$^{RNAi}$* antennae *Ir75a-GFP* expression is greatly diminished, while *Ir75b-GFP* is expressed in many ectopic cells (*Figure 6B–D*).

We further used these *Ir-GFP* transgenes to examine the glomerular projection of Ir75a and Ir75b neurons in *pdm3$^{RNAi}$* animals. In controls, *Ir75a-GFP* and *Ir75b-GFP* label exclusively the DP1l and DL2d glomeruli, respectively (*Figure 6E*), concordant with previous analyses (*Prieto-Godino et al., 2017*; *Silbering et al., 2011*). In *pdm3$^{RNAi}$* animals the *Ir75a-GFP* DP1l signal is almost completely lost (*Figure 6E*), consistent with the reduction in reporter expression in antennae. By contrast, the *Ir75b-GFP* signal in the DL2d glomerulus is greatly intensified, and the glomerulus is enlarged (*Figure 6E*), presumably reflecting the increased number of Ir75b neurons (*Figure 6C–D*). The simplest interpretation of these observations is that in the absence of Pdm3, Ir75a neurons adopt Ir75b neuron fate including both receptor identity and glomerular innervation pattern.

While OSN fate is established in early pupae (*Barish and Volkan, 2015*; *Jefferis and Hummel, 2006*; *Yan et al., 2020*), the expression of Pdm3 in Ir75a neurons in adult antennae (*Figure 5A–B*) suggests that it has a persistent role in these cells. To test this idea, we used a temperature-sensitive repressor of Gal4 (Gal80$^{ts}$; active at 19°C, inactive at 29°C) to temporally control the onset of *peb>pdm3$^{RNAi}$*. In control conditions, when we continuously suppressed (19°C) or permitted (29°C) *peb>pdm3$^{RNAi}$*, we reproduced our initial observations (*Figure 6F*, compare to *Figure 5D*). When *pdm3$^{RNAi}$* was induced only in adults (through a 19°C to 29°C temperature shift after eclosion) the same increase in Ir75b expression and decrease in Ir75a expression was observed (*Figure 6F*). Notably, we detected several cells expressing both IR75b and IR75a, which are never observed in non-RNAi antennae (*Figure 6G–H*). These observations indicate that Pdm3 has a persistent role in mature Ir75a neurons to repress *Ir75b* expression and promote *Ir75a* expression. Qualitatively similar, although much weaker, changes in receptor expression were observed when we drove *pdm3$^{RNAi}$* using *Ir8a-Gal4* (*Abuin et al., 2011*), which is active only from the second half of pupal development (~48 hr APF) (*Figure 6—figure supplement 1*). The ectopic expression of *Ir75b* in Ir75a OSNs was confirmed by examining the projection of *Ir75b-GFP*-expressing neurons: in adult-only *pdm3$^{RNAi}$* animals – but not in control pan-developmental *pdm3$^{RNAi}$* – *Ir75b-GFP* signal was detected both in

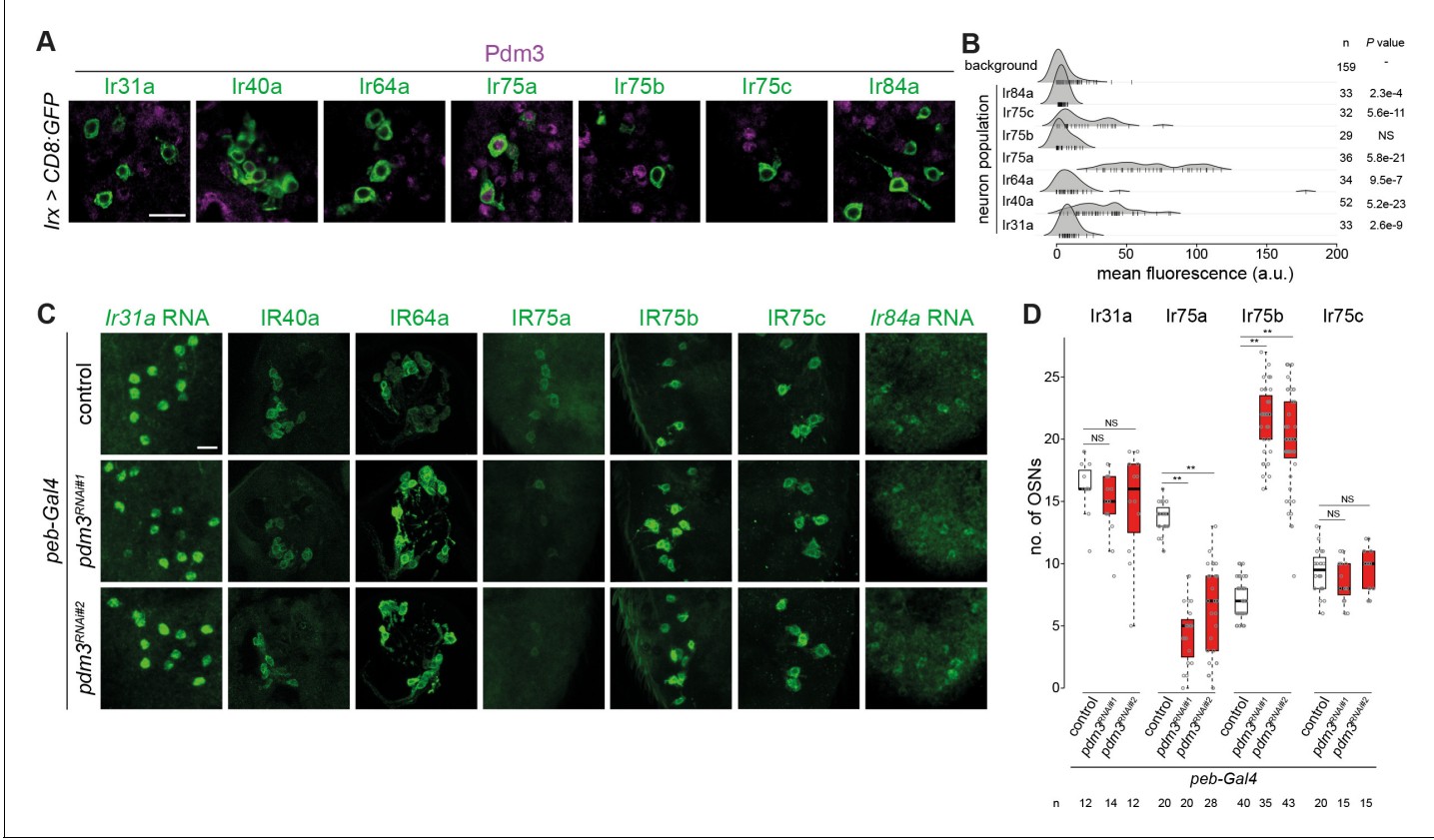

**Figure 5.** Heterogeneous expression and function of Pdm3 in Ir neurons. (A) Immunofluorescence for Pdm3 and GFP on antennal sections of animals in which the indicated Ir neuron populations are labeled with a CD8:GFP reporter. Genotypes are of the form: *Irxx-Gal4/+;UAS-mCD8:GFP/+* or, for Ir75a and Ir75c neurons, *UAS-mCD8:GFP/+;Irxx-Gal4/+*. Scale bar = 10 µm. (B) Density plots for Pdm3 immunofluorescence signals (arbitrary units, a.u.) quantified from Ir neuron nuclei (each represented by a vertical line) in the genotypes in (A) and from background signals pooled across genotypes (see Materials and methods). Ir75a neuron nuclei have the highest levels of immunofluorescence. n = sample size; *p*-value based on one-sided Wilcoxon tests, with Bonferroni correction for multiple comparisons. (C) Immunofluorescence or RNA FISH for the indicated IRs on whole-mount antennae (or antennal sections for IR40a and IR64a, due to poor antibody penetration of whole-mount tissue) of control animals (*peb-Gal4,UAS-Dcr-2/+;+/CyO*) or two independent *pdm3* RNAi lines (*peb-Gal4,UAS-Dcr-2/+;;UAS-pdm3^JF02312/+* (RNAi#1) and *peb-Gal4,UAS-Dcr-2/+;UAS-pdm3^HMJ21205/+* (RNAi#2)). Scale bar = 10 µm. (D) Quantification of neuron number for the indicated Ir neuron populations for the genotypes shown in (C). In this and other panels, boxplots show the median, first and third quartile of the data, overlaid with individual data points. Comparisons to controls are shown for each neuron type (pairwise Wilcoxon rank-sum two-tailed test and *p*-values adjusted for multiple comparisons with the Bonferroni method, **p<0.001; NS p>0.05). Sample sizes are shown below the plot. Ir40a and Ir64a OSN population sizes could not be quantified because the sections visualized necessarily contain a variable number of neurons; similarly, we could not confidently count Ir84a neuron number because of the weak signal. Nevertheless, when visualized blindly, the control and RNAi genotypes were not distinguishable for any of these neuron populations (assessing the phenotype in antennae of at least 10 animals from two independent genetic crosses).

DL2d and DP1l neurons (*Figure 6I*). This observation is consistent with late loss of *pdm3* leading to misexpression of *Ir75b-GFP* in Ir75a neurons after they have targeted to DP1l.

## Dynamic, heterogeneous expression of Fmi in OSNs

*fmi* attracted our attention both for its heterogeneous occupancy across Ir neuron populations (*Figure 4D–E*), and because this gene encodes an atypical cadherin – comprising an extracellular cadherin-like domain and a GPCR-like domain – with diverse roles in axonal and dendritic targeting in other regions of the nervous system (*Berger-Müller and Suzuki, 2011*; *Chen and Clandinin, 2008*; *Gao et al., 2000*; *Lee et al., 2003*; *Matsubara et al., 2011*; *Schwabe et al., 2013*). To characterize the endogenous expression of Fmi in the olfactory system, we used an antibody to profile protein levels in OSNs. We first examined antennae from mid-pupal stages (48 hr APF) but observed only trace levels of immunoreactivity across OSN soma (*Figure 7—figure supplement 1A*). At 24 hr APF, this signal was slightly stronger, but much more prominent in the OSN axons as they left the

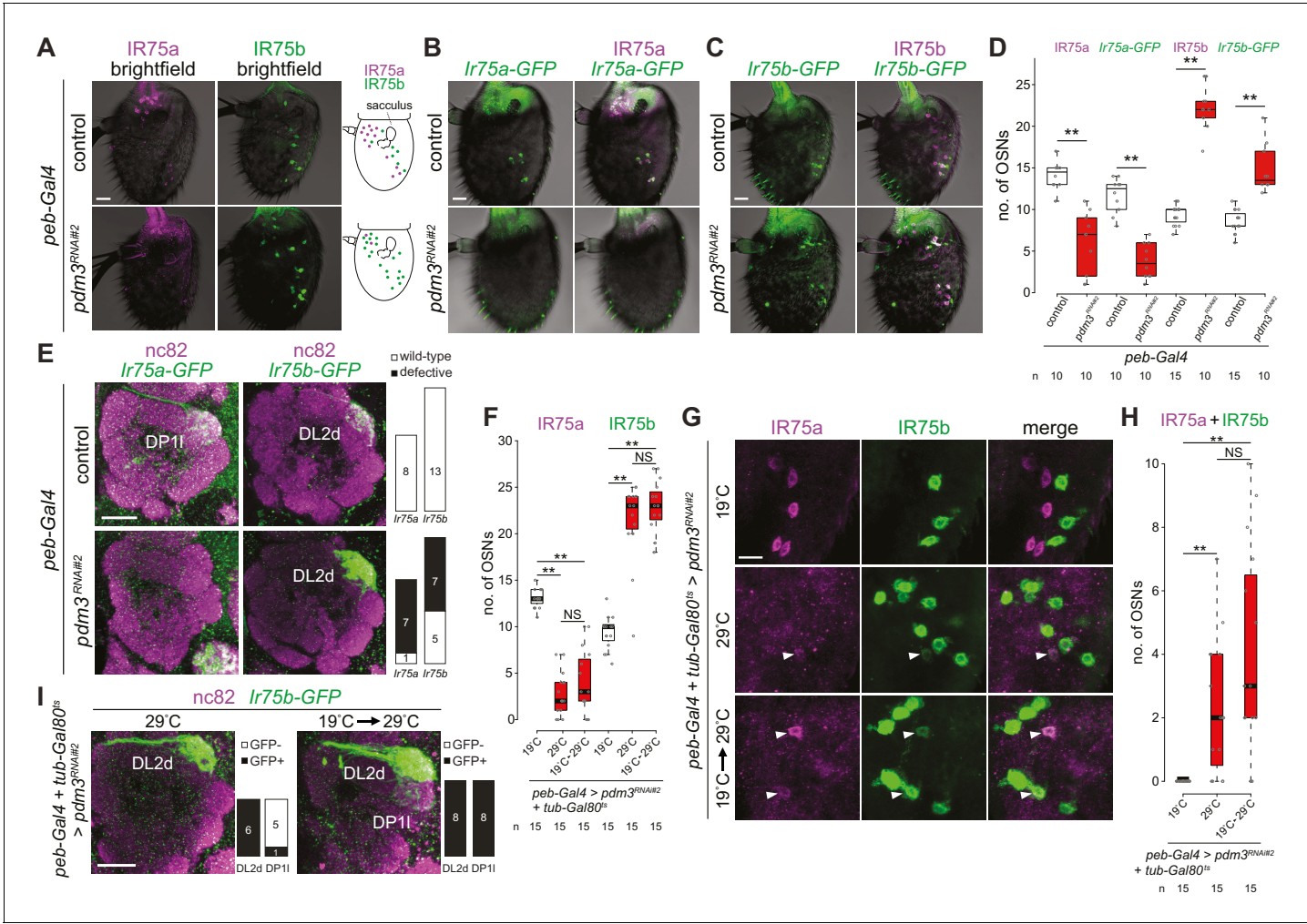

**Figure 6.** Pdm3 is required to distinguish Ir75a and Ir75b neuron fate. (**A**) Immunofluorescence for IR75a and IR75b on whole-mount antennae of control (*peb-Gal4,UAS-Dcr-2/+*) and *pdm3*[RNAi#2] (*peb-Gal4,UAS-Dcr-2/+;UAS-pdm3*[HMJ21205]/+) animals. Scale bar = 10 μm. The schematics on the right summarize the distribution of labeled neurons. (**B**) Immunofluorescence for GFP and IR75a on whole-mount antennae of control (*peb-Gal4,UAS-Dcr-2/+;Ir75a-GFP/+*) and *pdm3*[RNAi#2] (*peb-Gal4,UAS-Dcr-2/+;Ir75a-GFP/UAS-pdm3*[HMJ21205]) animals. Scale bar = 10 μm. (**C**) Immunofluorescence for GFP and IR75b on whole-mount antennae of control (*peb-Gal4,UAS-Dcr-2/+;Ir75b-GFP/+*) and *pdm3*[RNAi#2] (*peb-Gal4,UAS-Dcr-2/+;Ir75b-GFP/UAS-pdm3*[HMJ21205]) animals. Scale bar = 10 μm. (**D**) Quantification of the number of neurons that express *Ir75a-GFP* or *Ir75b-GFP* in the genotypes shown in (**B–C**). Comparisons to the controls are shown (pairwise Wilcoxon rank-sum two-tailed test and *p*-values adjusted for multiple comparisons with the Bonferroni method, \*\*p<0.001). The increase in number of *Ir75b-GFP* labeled neurons in *pdm3*[RNAi] is lower than the increase in IR75b-expressing neurons, potentially because the transgenic reporter is not fully faithful in this genetic background. (**E**) Immunofluorescence for nc82 and GFP on whole-mount antennal lobes of control and *pdm3*[RNAi#2] animals. Genotypes are as in (**B–C**). Scale bar = 20 μm. Quantification of phenotypes are shown on the right. (**F**) Quantification of the number of Ir75a and Ir75b neurons in animals (*peb-Gal4,UAS-Dcr-2/+;pdm3*[RNAi#2]/+;UAS-Gal80[ts]/+) in which *peb-Gal4*-driven *pdm3*[RNAi] is continuously suppressed (19°C, the permissive temperature for the Gal4 inhibitor Gal80[ts]), continuously allowed (29°C, the restrictive temperature for Gal80[ts]) or induced only in adults (19°C → 29°C temperature shift after eclosion). Comparisons between conditions are shown for each neuron type (pairwise Wilcoxon rank-sum two-tailed test with Bonferroni correction for multiple comparisons, \*\*p<0.001, NS p>0.05). (**G**) Immunofluorescence for IR75a and IR75b on whole-mount antennae of the genotypes shown in (**F**). Single optical sections are shown, to reveal the weak co-expression of IR75a and IR75b in a subset of cells (arrowheads). Scale bar = 10 μm. (**H**) Quantification of the number of neurons that co-express IR75a and IR75b. Comparisons to the controls are shown (pairwise Wilcoxon rank-sum two-tailed test with Bonferroni correction for multiple comparisons, \*\*p<0.001, NS p>0.05). (**I**) Immunofluorescence for nc82 and GFP on whole-mount antennal lobes of *pdm3*[RNAi#2] animals (*peb-Gal4,UAS-Dcr-2/+;UAS-pdm3*[HMJ21205]/Ir75b-GFP;UAS-Gal80[ts]/+) in which RNAi is allowed throughout development (29°C) or limited only to adults (19°C → 29°C temperature shift after eclosion). Quantification of glomerular labeling pattern by the *Ir75b-GFP* reporter is shown on the right of each image.

The online version of this article includes the following figure supplement(s) for figure 6:

**Figure supplement 1.** Phenotypic analysis of Ir75a and Ir75b neurons with late developmental induction of *pdm3* RNAi.

antenna (*Figure 7—figure supplement 1A*), suggesting that this protein is predominantly transported to the axonal compartment of these neurons. Indeed, Fmi protein was strongly detected in the OSN axons as they enter the antennal lobe at 24 hr APF (*Figure 7A*). Fmi was still robustly expressed at 48 hr APF as OSNs coalesced onto individual glomeruli with broad, but slightly heterogeneous, expression in glomeruli across the antennal lobe (*Figure 7A*). Subsequently, Fmi levels began to decrease to undetectable levels in some glomeruli in the adult stage, while persisting in others (*Figure 7B*). Most of these glomeruli are innervated by Or neurons, but we could detect Fmi in DL2d (Ir75b) and DL2v (Ir75c), but not the neighboring DP1l (Ir75a) (*Figure 7B*), in partial concordance with the TaDa data (*Figure 4E*).

From later developmental stages (~48 hr APF), Fmi was also detected in a group of ~16 cells located on the antero-dorsal region of the antennal lobe (*Figure 7A*). These cells are likely to be LNs, based upon their expression of Elav (*Figure 7—figure supplement 1B*) and presence of stained processes that innervate the antennal lobe but do not project to higher olfactory centers. Numerous LN subtypes exist, and these often have broad innervation patterns across many glomeruli (*Chou et al., 2010a*; *Liou et al., 2018*). To examine the specific contribution of OSNs to the Fmi immunoreactivity observed in the antennal lobe, we depleted Fmi in LNs by RNAi using a combination of Gal4 drivers (*1081* Gal4 and *449* Gal4 [*Liou et al., 2018*]) that cover the vast majority of these interneurons (*Figure 7—figure supplement 1C*). At mid-pupal development (48 hr APF), differential glomerular expression of Fmi was much more marked compared to control animals (*Figure 7C*). Although we could not confidently define all glomerular identities at this earlier developmental stage, prior to full maturation of the antennal lobe, Fmi protein levels did not appear to correspond precisely with the TaDa OSs (e.g. the VL2a glomerulus, innervated by Ir84a neurons, was not devoid of Fmi signal). This observation suggests that Fmi is dynamically expressed in different populations of OSNs during development. We note that the protein observed at 48 hr APF must be due to transcription of *fmi* at an earlier time point than captured by our TaDa analysis. It is also possible that different post-transcriptional regulation within OSN populations contributes to lack of strict correlation between transcript and protein levels with these cells, as noted in other tissues (*Liu et al., 2016*). Nevertheless, this analysis reveals developmentally dynamic and population-specific control of Fmi expression in OSNs.

## Fmi is required for segregation of OSNs into distinct glomeruli

To determine the function of Fmi in the developing olfactory system, we induced *fmi* RNAi in developing antennal tissue first using a constitutive driver (*ey-Flp,act>stop>Gal4* [*Chai et al., 2019*]), which should deplete Fmi expression from the earliest stages of development, and examined adult brains stained with the general neuropil marker nc82 (Bruchpilot). The antennal lobes of these animals showed fully penetrant, severe morphological defects: the stereotyped glomerular boundaries observed in the lobes of control animals were mostly lacking, with only a few, large glomerulus-like subregions detected (*Figure 8A*). This phenotype was reproduced with an independent RNAi line that targets a distinct region of the coding sequence (*Figure 8A*).

We further examined this phenotype by visualizing the projection patterns of specific classes of OSNs labeled by receptor promoter-driven fluorescent reporters (*Figure 8B–C*). Neurons labeled by *Ir75a-GFP*, *Ir75b-GFP*, or *Ir75c-GFP* were no longer constrained to discrete glomeruli (*Figure 8B*), although the mistargeting was relatively mild within the antennal lobe as a whole. The limited defects of individual neuron populations suggests that the lack of Fmi does not disrupt long-range targeting properties of OSNs, but rather impacts local segregation of OSNs. Consistently, double labeling of neurons that normally project to adjacent glomeruli (*Ir75a-GFP* and *Ir75b-Tomato* (*Tom*); *Ir84a-GFP* and *Ir31a-Tom*) revealed frequent overlap between these normally-segregated populations (*Figure 8C*).

The constitutive driver is expressed broadly across the antennal disc (*Chai et al., 2019*). The observed phenotypes could therefore potentially be due to a requirement for Fmi in disc development rather than in OSNs. To limit *fmi* RNAi to OSNs, we used *peb-Gal4*, which produced an equally strong defect in antennal lobe glomerular segregation (*Figure 8D*). In these animals, the LN-expressed Fmi was still detectable but, in contrast to the expression in OSNs, Fmi was apparently homogeneously distributed across antennal lobe glomeruli (*Figure 7—figure supplement 1D*). This LN source of Fmi does not appear to contribute to glomerulus formation as *LN>fmi^(RNAi)* animals did not exhibit antennal lobe morphological defects (*Figure 8D*). Similarly, when we targeted *fmi* to the

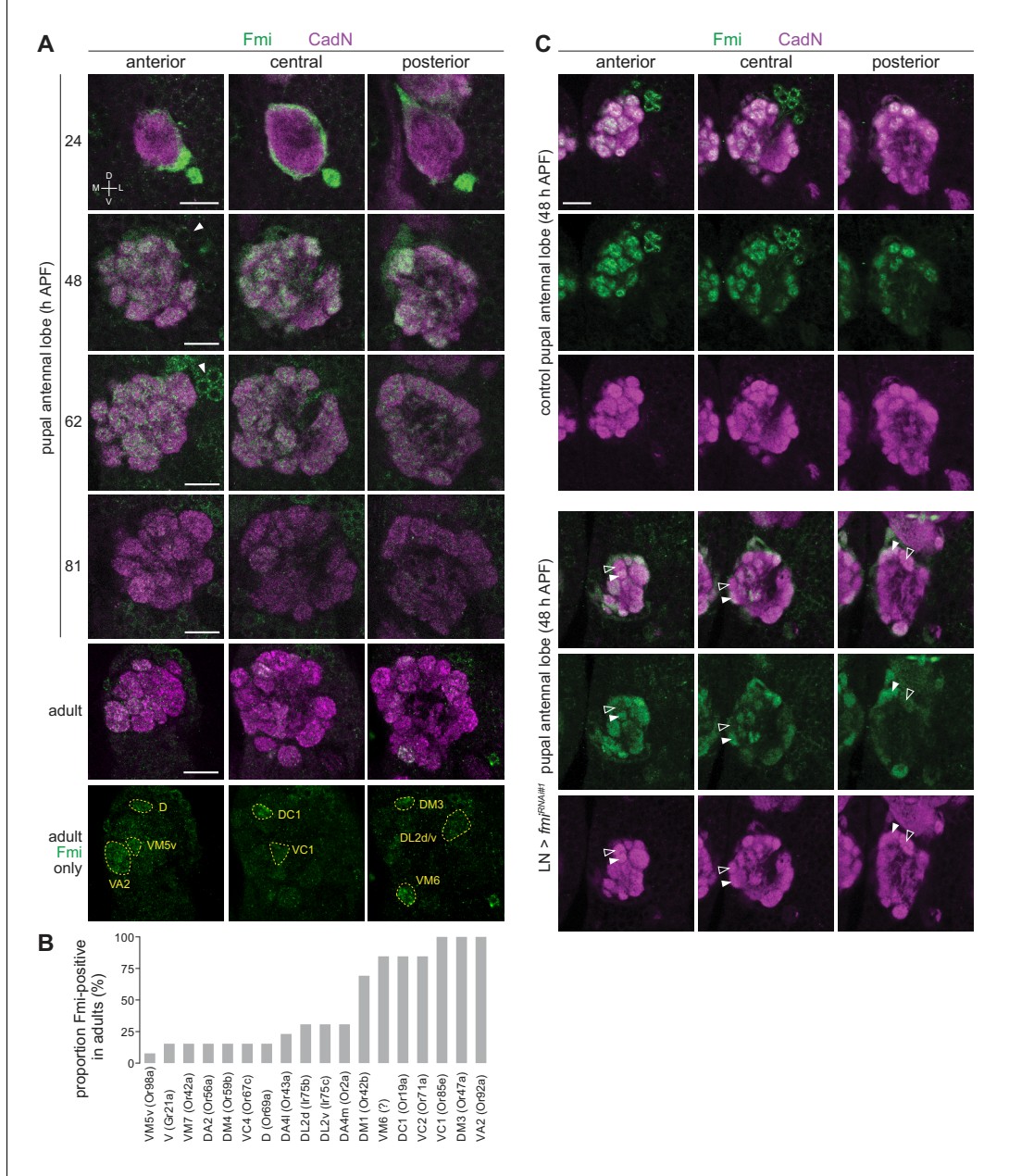

**Figure 7.** Expression analysis of Fmi in OSNs. (**A**) Immunofluorescence for Fmi and the ubiquitously-expressed neuronal cadherin (CadN) on whole-mount antennal lobes of wild-type ($w^{1118}$) animals of the indicated age. Three optical sections are shown to reveal the morphology of most glomeruli. Bottom row: Fmi-positive glomeruli were identified in adult antennal lobes based upon their stereotyped position and morphology. Dorsal-ventral (D-V) and medial-lateral (M-L) axes are indicated. The arrowheads point to the soma of Fmi-expressing LNs that are detectable from 48 hr APF. Scale bar = 20 µm. (**B**) Histogram of frequency of detectable Fmi immunoreactivity in individual glomeruli of the adult antennal lobe (n = 14 antennal lobes). (**C**) Immunofluorescence for Fmi and CadN on whole-mount antennal lobes of control (*1081-Gal4/+;449-Gal4/UAS-Dcr-2*) and *LN>fmi^{RNAi#1}* (*1081-Gal4/UAS-fmi^{KK100512};449-Gal4/UAS-Dcr-2*) 48 hr APF animals. Open and filled arrowheads highlight adjacent glomeruli with low and high levels, respectively, of Fmi immunoreactivity. Scale bar = 20 µm.

The online version of this article includes the following figure supplement(s) for figure 7:

**Figure supplement 1.** Characterization of Fmi expression in antennae and local interneurons.

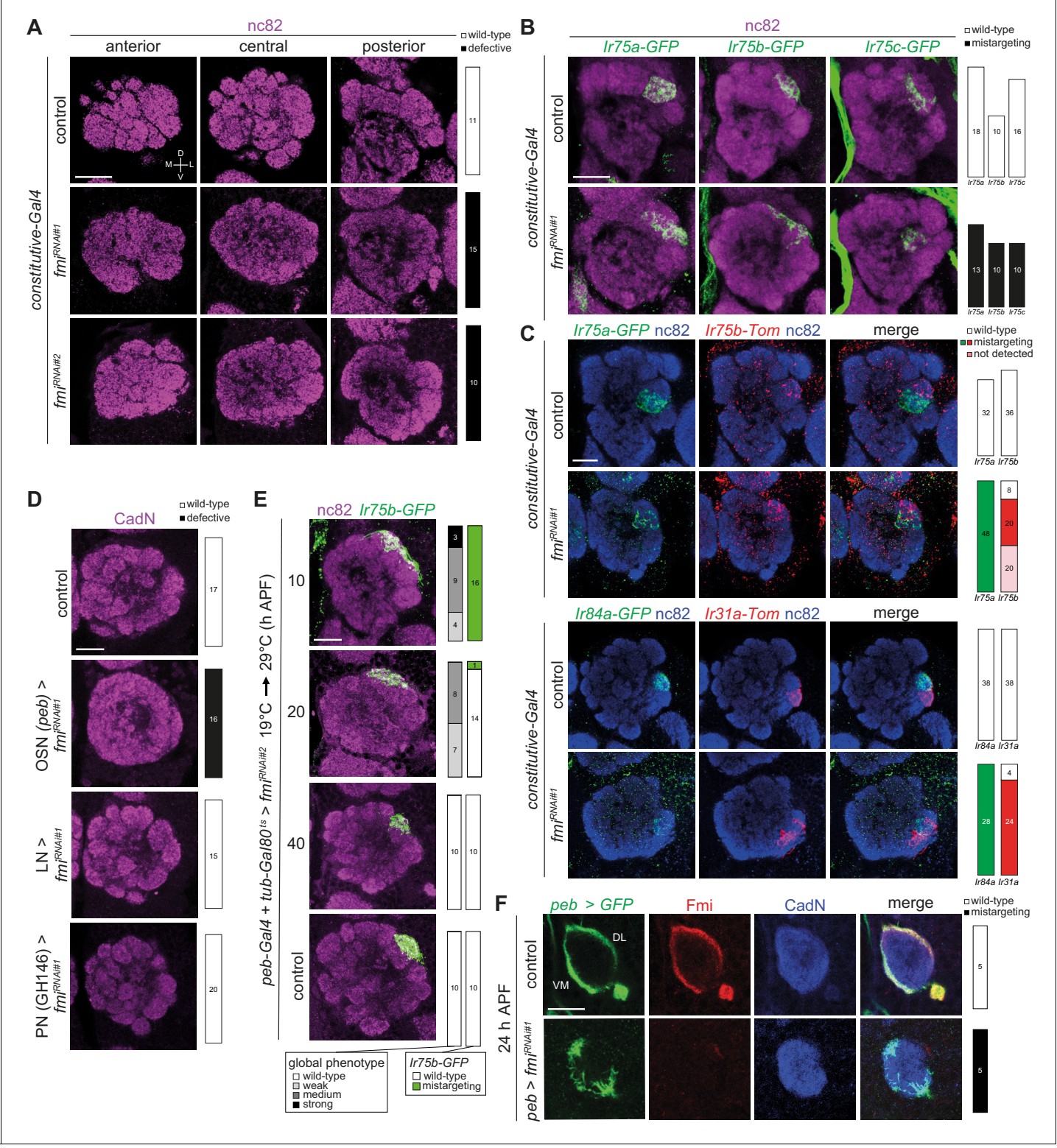

**Figure 8.** Fmi is required in OSNs for glomerular segregation. (**A**) Immunofluorescence for nc82 on whole-mount antennal lobes of control (*w;ey-Flp/+; act>stop>Gal4/+*), *fmi^{RNAi#1}* (*w;ey-Flp/UAS-fmi^{KK100512};act>stop>Gal4/+*) and *fmi^{RNAi#2}* (*w;ey-Flp/+;act>stop>Gal4/UAS-fmi^{JF02047}*) animals. Three optical sections are shown. Scale bar = 20 μm. Quantification of wild-type or defective antennal lobe glomerular architectures are shown on the right; the n for each phenotypic category are indicated on the plots. (**B**) Immunofluorescence for nc82 and GFP on whole-mount antennal lobes of control and *fmi^{RNAi#1}* animals. Genotypes: *Ir75a-GFP* (*w;ey-Flp,Ir75a-GFP/[+ or UAS-fmi^{KK100512}];act>stop>Gal4/+*), *Ir75b-GFP* (*w;ey-Flp,Ir75b-GFP/[+ or UAS-*
*Figure 8 continued on next page*

Figure 8 continued

$fmi^{KK100512}$];*act>stop>Gal4/+*), Ir75c-GFP (*w;ey-Flp,Ir75c-GFP/[+ or UAS-fmi^{KK100512}];act>stop>Gal4/+*). Scale bar = 20 μm. (**C**) Immunofluorescence for nc82, GFP, RFP (detects Tomato (Tom)) on whole-mount antennal lobes of control and $fmi^{RNAi#1}$ animals. Genotypes: *w;Ir75a-GFP,Ir75b-Tom/[+ or UAS-fmi^{KK100512}];ey-FLP,act>stop>Gal4/Ir75b-RFP* (top panels) and *eyFlp,act>stop>Gal4/[+ or UAS-fmi^{KK100512}];Ir31a-Tom,Ir84a-GFP/+* (bottom panels). Scale bar = 20 μm. Quantification of wild-type or defective targeting phenotypes are shown on the right, as indicated in the key at the top-right. (**D**) Immunofluorescence for CadN on whole-mount antennal lobes of control (*peb-Gal4*), OSN>$fmi^{RNAi#1}$ (*peb-Gal4/+;UAS-fmi^{KK100512}/+;UAS-Dcr-2/+*), LN>$fmi^{RNAi#1}$ (*1081-Gal4/UAS-fmi^{KK100512};449-Gal4/UAS-Dcr-2*), PN>$fmi^{RNAi#1}$ (*GH146-Gal4/+;UAS-fmi^{KK100512}/+*) animals. Scale bar = 20 μm. (**E**) Immunofluorescence for nc82 and GFP on whole-mount antennal lobes of *peb-Gal4,UAS-Dcr-2/+;Ir75b-GFP/tub-Gal80^{ts};UAS-fmi^{JF02047}/+* animals in which Gal4-driven RNAi was suppressed until shifting from the permissive to restrictive temperature (19°C → 29°C) for Gal80$^{ts}$ at the indicated time points. Scale bar = 20 μm. (**F**) Immunofluorescence for GFP, Fmi and CadN on whole-mount pupal antennal lobes (24 hr APF) of control (*peb-Gal4/+;;UAS-mCD8::GFP/+*) and OSN>$fmi^{RNAi#1}$ (*peb-Gal4/+;UAS-fmi^{KK100512}/+;UAS-mCD8::GFP/+*) animals. The merged channels are shown on the right. The dorsolateral (DL) and ventromedial (VM) bundles of OSN axons are indicated. Scale bar = 20 μm.

The online version of this article includes the following figure supplement(s) for figure 8:

**Figure supplement 1.** Fmi does not act in OSNs with known genetic and physical interaction partners.

synaptic partners of OSNs, the PNs – which pre-pattern the glomerular organization of the antennal lobe (*Jefferis and Hummel, 2006*) – using the *GH146* driver, antennal lobe morphology appeared normal (*Figure 8D*). The absence of defects is consistent with the lack of detectable expression of Fmi in these second-order neurons (*Figure 7A*). Together, these results indicate that Fmi functions principally, if not exclusively, within OSNs for correct glomerulus formation.

To understand the genesis of this defect during OSN development we used Gal80$^{ts}$ to temporally-control the onset of *peb>fmi$^{RNAi}$*. When animals were shifted from 19°C to 29°C at 10 hr APF (prior to initial expression of *peb-Gal4*), all animals displayed antennal lobe morphological defects of varying degrees, and *Ir75b-GFP*-labeled neurons did not coalesce to a discrete glomerular unit (*Figure 8E*). When shifted at 20 hr APF, the severity of the phenotype was diminished (*Figure 8E*). When shifted at 40 hr APF, the antennal lobe and Ir75b neuron targeting were indistinguishable from controls (*Figure 8E*). These experiments indicate an early developmental function for Fmi.

We examined this early requirement in more detail by imaging OSN axons as they invade the antennal lobe. During 20–25 hr APF, pioneer OSNs reach the antennal lobe in a unique bundle that separates into ventromedial and dorsolateral bundles, which project around the lobe's border (*Hong and Luo, 2014*; *Joo et al., 2013*; *Figure 8F*). In *peb>fmi$^{RNAi}$* animals, OSNs reach the antennal lobe, but bundle organization is highly disrupted, with subsets of axons apparently defasciculating prematurely in the central area of the lobe (*Figure 8F*).

Fmi (and its mammalian homologs) have roles in multiple biological processes in the nervous system as well as in the establishment of epithelial planar cell polarity (*Berger-Müller and Suzuki, 2011*; *Keeler et al., 2015*; *Takeichi, 2007*). Genetic and biochemical studies have identified a number of interaction partners of Fmi in different biological contexts. We tested mutations and/or multiple independent RNAi lines for several of these genes to determine whether they function with this cadherin in OSNs (*Figure 8—figure supplement 1*). Mutations in two core components of the planar cell polarity pathway, *frizzled* and *dishevelled* (*Usui et al., 1999*), had no apparent impact on antennal lobe formation. Similarly, loss of *frazzled* (encoding a Netrin homolog which function with Fmi in midline axon guidance [*Organisti et al., 2015*]), *golden goal* (with which Fmi collaborates in photoreceptor axon guidance [*Hakeda-Suzuki et al., 2011*]), or *espinas* (which encodes a LIM domain protein that binds Fmi intracellularly and functions in dendrite self-avoidance [*Matsubara et al., 2011*]) did not reproduce the *fmi* phenotype. Finally, although the intracellular signaling mechanism of Fmi is not well understood, the Fmi GPCR-domain has been suggested to act through the G$\alpha_q$ subunit to regulate dendrite growth (*Wang et al., 2016*); however, we found no evidence that this signaling mechanism operates in OSNs (*Figure 8—figure supplement 1*). These observations suggest that Fmi functions through other cellular mechanisms to control OSN axon segregation.

## Discussion

Determination of the molecular composition of neurons is essential to understand the development and function of the nervous system. While scRNA sequencing is an undoubtedly powerful approach to address this challenge (*Croset et al., 2018*; *Davie et al., 2018*; *Li et al., 2017*; *Tosches et al.,*

*2018*; *Zeisel et al., 2018*), it requires often technically difficult cell isolation, and can be biased toward abundant cell types and highly expressed genes (*Stegle et al., 2015*). Moreover, it is unclear to what extent neurons change their expression properties during tissue dissection (as reported for other types of cell [*van den Brink et al., 2017*]) or through inevitable loss of dendritic and axonal processes (where certain transcripts may be enriched [*Middleton et al., 2019*]). Our study was initially motivated by the desire to test the TaDa method for targeted, genome-wide molecular profiling of very small populations of neurons that are tightly embedded within a highly heterogeneous tissue. Moreover, by applying TaDa to a set of functionallyrelated OSNs, we aimed to identify cell type-specific differences that would point toward mechanisms underlying the development and evolution of novel neuronal populations.

The unique olfactory receptor expression pattern of OSNs (and other chemosensory genes) offered several positive- and negative-control loci that allowed us to validate the specificity and sensitivity of TaDa. One caveat emerging from this analysis is that TaDa cannot easily discriminate PolII occupancy of interleaved genes, an issue that may be particularly pronounced in the relatively gene-dense genome of *Drosophila*. Nevertheless, our results indicate that this method could be effective in defining candidate receptors in neuronal classes for which a genetic driver is available. This possibility is of particular interest in the *Drosophila* gustatory system where many populations of neurons can be labeled and physiologically profiled (*Chen and Dahanukar, 2020*), but incomplete knowledge of the combinatorial receptor expression properties that are characteristic of this sensory system hampers identification of the cognate sensory receptor(s) (*Croset et al., 2016*; *Lee et al., 2018*; *Sánchez-Alcañiz et al., 2018*).

Recent work described a scRNA-seq analysis of antennal OSNs at a mid-pupal stage (42–48 hr APF) (*Li et al., 2020*), and a preprint has reported pioneering RNA-seq profiling of single nuclei (snRNA-seq) of 24 hr APF and adult stage OSNs (*McLaughlin et al., 2020*). Together these RNA-seq studies identified transcriptional clusters for 26–34 types of OSNs (depending upon the stage); of these, 16 molecularly annotated classes (i.e., expressing specific receptors) could be recognized at all three developmental timepoints. Detailed comparison of these scRNA-seq and snRNA-seq data and our TaDa datasets is currently difficult, as there is only limited overlap of the identified sc/snRNA-seq clusters and the populations we analyzed by TaDa. Notably, the adult OSN snRNA-seq datasets clustered together neurons expressing *Ir75a*, *Ir75b* and *Ir75c* (as well as Or35a neurons). This highlights one advantage of our targeted approach in selectively distinguishing these Ir populations, which enabled our discovery of the Ir75a neuron-specific expression and function of Pdm3. It is important to appreciate that while the sc/snRNA-seq captures the transcriptional state during a relatively narrow developmental window, our TaDa experiments integrated PolII occupancy patterns at an OSN population level over several days from the initiation of *Ir-Gal4* expression at mid-pupal development. This difference in temporal profiling might explain, in part, the higher numbers of genes detected by TaDa (on average ~4000 per Ir population) compared to pupal OSN scRNA-seq (~1500 expressed genes per cell [*Li et al., 2020*]) and adult OSN snRNA-seq (~1100 genes per nucleus [*McLaughlin et al., 2020*]). However, many other experimental and/or analytical differences in these methods are likely to impact such global numbers. At the level of individual genes, one intriguing difference between our studies is the broad neuronal expression of several *Obp*s in the snRNA-seq data (*McLaughlin et al., 2020*) compared to the extremely sparse PolII occupancy of *Obp* genes in our TaDa datasets. All prior in situ RNA and protein expression analyses of OBPs in *Drosophila* and other insect species have recognized their expression in support cells but not in neurons (*Sun et al., 2018b*). It is possible that the snRNA-seq analyses has higher sensitivity than in situ and TaDa analyses and therefore captured previously unappreciated neuronal expression. The function of these OBPs in OSNs is, however, unclear.

Genome-wide analysis of TaDa datasets from different Ir populations revealed an intriguing positive relationship between the similarity of global patterns of PolII occupancy between neuronal populations and the phylogenetic distance of the olfactory receptors they express, notably the tandem array of *Ir75a*, *Ir75b*, and *Ir75c* genes. Similar analysis of Or OSN populations is limited by the broad phylogenetic distribution of the ORs whose corresponding neuronal transcriptomes have been identified (*Li et al., 2020*; *McLaughlin et al., 2020*). However, through examination of these data (*Li et al., 2020*), we identified two pairs of closely-related (though not tandemly-arranged) *Or* genes whose neuronal transcriptional profiles also cluster (*Or42b/Or59b* and *Or9a/Or47a*). The simplest interpretation of these observations is that neuronal populations expressing distinct, relatively-recent

receptor gene duplicates derive from a common ancestral neuronal population (which may have originally co-expressed the duplicated receptors); these extant populations therefore retain correspondingly close proximity in their transcriptomic profile. The principle of OSN 'sub-functionalization' – which has not, to our knowledge, previously been recognized empirically – has interesting implications for understanding the evolution of novel olfactory pathways (*Ramdya and Benton, 2010*).

One advantage of the TaDa profiling method over scRNA-seq is that it can also provide information on chromatin state, which remains difficult to characterize through standard chromatin immunoprecipitation-based methods (*Ludwig and Bintu, 2019*). We recognize that CATaDa provides only a relatively coarse-grained view of chromatin state (similar to ATAC-seq [*Aughey et al., 2018*]), and variant approaches of TaDa (e.g. Dam fusions to chromatin-binding proteins [*van den Ameele et al., 2019*]) are very likely to reveal finer-scale differences between populations. Nevertheless, our findings of globally similar chromatin accessibility at *Ir* genes across Ir populations are in-line with studies in the developing *Drosophila* embryo where chromatin accessibility is comparable between anterior and posterior region, despite substantial differences in patterns of gene transcription across this body axis (*Haines and Eisen, 2018*). Although we did identify a few regions of chromatin accessibility in receptor gene regulatory regions that are specific to their corresponding neuronal population, we suggest that the unique developmental properties of OSNs are driven, in larger part, by the particular combinations of TFs they express. Indeed, many of the differentially occupied genes between Ir populations encode known or predicted TFs. We validated the selective expression and function of Pdm3 in Ir75a neurons, revealing it to be a key factor – and potentially the sole transcription factor – that distinguishes the fate of closely related Ir75a and Ir75b neuron populations, encompassing both their receptor expression and glomerular targeting properties. Beyond this early developmental role, Pdm3 expression persists in Ir75a neurons into adulthood, and we showed it has an ongoing role to maintain *Ir75b* in a repressed state and promote *Ir75a* expression. Future identification of Pdm3 targets in these neurons will be necessary to determine whether it regulates these *Ir*s directly, and to identify other downstream genes that control the innervation of the corresponding neurons to distinct glomeruli.

Our TaDa datasets also suggest a rich diversity in the repertoires of signaling and neuronal morphogenesis molecules expressed in Ir neurons. Because of the temporal breadth of TaDa profiling, these proteins are likely to contribute to differences of these neurons in cilia/dendritic morphogenesis, axon targeting, sensory transduction, synapse formation/plasticity and/or neuromodulation. Our initial characterization of one of these, the atypical cadherin Fmi, reveal it to be a dynamically and heterogeneously expressed, axon-localized protein. Importantly, Fmi is not restricted to Ir neurons, but found throughout the peripheral olfactory system, concordant with the dramatic loss of glomerular architecture in the antennal lobe in the absence of this protein. We have delimited the role of Fmi specifically to OSNs as they first defasciculate and enter the antennal lobe. Differential expression of Fmi has been proposed to be important for its role in photoreceptor axon guidance (*Chen and Clandinin, 2008*). As Fmi is thought to function as a homophilic adhesion molecule (*Usui et al., 1999*), we speculate that heterogeneous Fmi-dependent adhesion between OSN populations enables them to segregate within the axon bundle and/or defasciculate at specific times and places, before local, instructive guidance cues direct them to particular locations within the antennal lobe. Unfortunately, the early requirement for Fmi and the inability to discern levels of Fmi in specific OSN populations within the bundle makes it difficult to assess the significance of the heterogeneous expression of this cadherin. We suspect that our pan-OSN *fmi^{RNAi}* has both direct effects on neurons that express higher levels of this protein, and indirect effects on those with low or absent Fmi expression. This important issue will require removal of Fmi by RNAi using by early drivers for specific populations of OSNs (which are as-yet unavailable) or clonal analysis of loss-of-function mutations of *fmi*. We have attempted to overexpress Fmi to assess the consequence of this alternative perturbation of Fmi levels on OSN axonal projections but this did not yield overt defects (data not shown). This negative result may simply reflect technical limitation in achieving adequate high-level Fmi overexpression, as reported in other systems (*Gao et al., 2000*). An adhesion-dependent role for Fmi is consistent with the *fmi* phenotype most closelyresembling that of mutations in *CadN* (*Hummel and Zipursky, 2004*). CadN is however homogeneously expressed across the antennal lobe, and also present in their PN synaptic partners (*Hummel and Zipursky, 2004*; *Zhu and Luo, 2004*). Future cell typespecific expression and functional manipulation will be required to understand

the special role of the atypical cadherin Fmi in correct segregation of OSNs axons to define discrete glomeruli.

Beyond further functional characterization of differentially occupied genes identified in this study, several future applications of TaDa (and CATaDa) in the olfactory system, and other sensory systems, can be envisaged. For example, temporal analysis of PolII occupancy could be refined by restricting Gal4 activity (with Gal80$^{ts}$) to a particular developmental time-window. Mining of our TaDa datasets for population-specific genes may help to design novel OSN lineage-specific driver lines that are expressed earlier in development (of which few are currently known [*Chai et al., 2019*]) to profile gene expression patterns prior to onset of receptor expression. Finally, in adult flies, comparison of PolII occupancy patterns in a specific neuron population before, during and after odor exposure could provide insight into neural activity-regulated gene expression and its relevance for sensory adaptation and plasticity.

# Materials and methods

## Key resources table

| Reagent type (species) or resource | Designation | Source or reference | Identifiers | Additional information |
|---|---|---|---|---|
| Genetic reagent (*D. melanogaster*) | *w$^{1118}$* | Bloomington *Drosophila* Stock Center | RRID:BDSC_3605 | |
| Genetic reagent (*D. melanogaster*) | *Ir8a-Gal4* | Bloomington *Drosophila* Stock Center | RRID:BDSC_41731 | |
| Genetic reagent (*D. melanogaster*) | *Ir31a-Gal4* | Bloomington *Drosophila* Stock Center | RRID:BDSC_41726 | |
| Genetic reagent (*D. melanogaster*) | *Ir40a-Gal4* | Bloomington *Drosophila* Stock Center | RRID:BDSC_41727 | |
| Genetic reagent (*D. melanogaster*) | *Ir64a-Gal4* | Bloomington *Drosophila* Stock Center | RRID:BDSC_41732 | |
| Genetic reagent (*D. melanogaster*) | *Ir75a-Gal4* | Bloomington *Drosophila* Stock Center | RRID:BDSC_41748 | |
| Genetic reagent (*D. melanogaster*) | *Ir75b-Gal4* | *Prieto-Godino et al., 2017* | | |
| Genetic reagent (*D. melanogaster*) | *Ir75c-Gal4* | *Prieto-Godino et al., 2017* | | |
| Genetic reagent (*D. melanogaster*) | *Ir84a-Gal4* | Bloomington *Drosophila* Stock Center | RRID:BDSC_41734 | |
| Genetic reagent (*D. melanogaster*) | *peb-Gal4* | Bloomington *Drosophila* Stock Center | RRID:BDSC_80570 | |
| Genetic reagent (*D. melanogaster*) | *ey-Flp (II)* | Bloomington *Drosophila* Stock Center | RRID:BDSC_5576 | |
| Genetic reagent (*D. melanogaster*) | *ey-Flp (III)* | Bloomington *Drosophila* Stock Center | RRID:BDSC_5577 | |
| Genetic reagent (*D. melanogaster*) | *act>stop>Gal4 (II)* | Bloomington *Drosophila* Stock Center | RRID:BDSC_3953 | |

*Continued on next page*

*Continued*

| Reagent type (species) or resource | Designation | Source or reference | Identifiers | Additional information |
|---|---|---|---|---|
| Genetic reagent (*D. melanogaster*) | *act>stop>Gal4 (III)* | Bloomington *Drosophila* Stock Center | RRID:BDSC_4780 | |
| Genetic reagent (*D. melanogaster*) | *GH146-Gal4* | Bloomington *Drosophila* Stock Center | RRID:BDSC_30026 | |
| Genetic reagent (*D. melanogaster*) | *449-Gal4* | *Liou et al., 2018* | | |
| Genetic reagent (*D. melanogaster*) | *1081-Gal4* | *Liou et al., 2018* | | |
| Genetic reagent (*D. melanogaster*) | *elav-Gal4* | Bloomington *Drosophila* Stock Center | RRID:BDSC_458 | |
| Genetic reagent (*D. melanogaster*) | *tub-Gal80$^{ts}$* | Bloomington *Drosophila* Stock Center | RRID:BDSC_7018 | |
| Genetic reagent (*D. melanogaster*) | *UAS-Dcr-2 (X)* | Bloomington *Drosophila* Stock Center | RRID:BDSC_24648 | |
| Genetic reagent (*D. melanogaster*) | *UAS-Dcr-2 (II)* | Bloomington *Drosophila* Stock Center | RRID:BDSC_24650 | |
| Genetic reagent (*D. melanogaster*) | *UAS-Dcr-2 (III)* | Bloomington *Drosophila* Stock Center | RRID:BDSC_24651 | |
| Genetic reagent (*D. melanogaster*) | *UAS-LT3-Dam (III)* | *Southall et al., 2013* | | |
| Genetic reagent (*D. melanogaster*) | *UAS-LT3-Dam:RpII15 (III)* | *Southall et al., 2013* | | |
| Genetic reagent (*D. melanogaster*) | *UAS-mCD8:GFP* | Bloomington *Drosophila* Stock Center | RRID:BDSC_5130 | |
| Genetic reagent (*D. melanogaster*) | *UAS-RedStinger* | Bloomington *Drosophila* Stock Center | RRID:BDSC_8546 | |
| Genetic reagent (*D. melanogaster*) | *Ir75b-CD4:tdGFP* | *Prieto-Godino et al., 2017* | | |
| Genetic reagent (*D. melanogaster*) | *UAS-fmi$^{KK100512}$* (RNAi #1) | Vienna *Drosophila* Resource Center | VDRC v107993 | |
| Genetic reagent (*D. melanogaster*) | *UAS-fmi$^{JF02047}$* (RNAi #2) | Bloomington *Drosophila* Stock Center | RRID:BDSC_26022 | |
| Genetic reagent (*D. melanogaster*) | *UAS-fra$^{HMS01147}$* (RNAi) | Bloomington *Drosophila* Stock Center | RRID:BDSC_40826 | |
| Genetic reagent (*D. melanogaster*) | *UAS-fra$^{JF01231}$* (RNAi) | Bloomington *Drosophila* Stock Center | RRID:BDSC_31469 | |
| Genetic reagent (*D. melanogaster*) | *UAS-fra$^{JF01457}$* (RNAi) | Bloomington *Drosophila* Stock Center | RRID:BDSC_31664 | |
| Genetic reagent (*D. melanogaster*) | *UAS-Gaq$^{dsRNA.UAS.1f1}$* (RNAi) | Bloomington *Drosophila* Stock Center | RRID:BDSC_30735 | |

*Continued on next page*

*Continued*

| Reagent type (species) or resource | Designation | Source or reference | Identifiers | Additional information |
|---|---|---|---|---|
| Genetic reagent (*D. melanogaster*) | *UAS-Gaq*[JF01209] (RNAi) | Bloomington *Drosophila* Stock Center | RRID:BDSC_31268 | |
| Genetic reagent (*D. melanogaster*) | *UAS-Gaq*[GL01048] (RNAi) | Bloomington *Drosophila* Stock Center | RRID:BDSC_36820 | |
| Genetic reagent (*D. melanogaster*) | *UAS-gogo*[GD3616] (RNAi) | Vienna *Drosophila* Resource Center | VDRC v43928 | |
| Genetic reagent (*D. melanogaster*) | *UAS-gogo*[HMC05937] (RNAi) | Bloomington *Drosophila* Stock Center | RRID:BDSC_65193 | |
| Genetic reagent (*D. melanogaster*) | *UAS-pdm3*[JF02312] (RNAi #1) | Bloomington *Drosophila* Stock Center | RRID:BDSC_26749 | |
| Genetic reagent (*D. melanogaster*) | *UAS-pdm3*[HMJ21205] (RNAi #2) | Bloomington *Drosophila* Stock Center | RRID:BDSC_53887 | |
| Genetic reagent (*D. melanogaster*) | *dsh*[1] | *Krasnow et al., 1995* | | |
| Genetic reagent (*D. melanogaster*) | *esn*[KO6] | *Matsubara et al., 2011* | | |
| Genetic reagent (*D. melanogaster*) | *fz*[H51] | *Jones et al., 1996* | | |
| Genetic reagent (*D. melanogaster*) | *fz*[KD4a] | *Adler et al., 1990* | | |
| Genetic reagent (*D. melanogaster*) | *fz*[P21] | *Jones et al., 1996* | | |
| Genetic reagent (*D. melanogaster*) | *Ir31a-CD4:tdTom* | This work | | |
| Genetic reagent (*D. melanogaster*) | *Ir75a-CD4:tdGFP* | This work | | |
| Genetic reagent (*D. melanogaster*) | *Ir75b-CD4:tdTom* | This work | | |
| Genetic reagent (*D. melanogaster*) | *Ir75c-CD4:tdGFP* | This work | | |
| Genetic reagent (*D. melanogaster*) | *Ir84a-CD4:tdGFP* | This work | | |
| Antibody | Anti-En (mouse monoclonal) | DSHB 4D9 | RRID:AB_528224 | (1:100) |
| Antibody | Anti-Pdm3 (guinea pig polyclonal) | *Chen et al., 2012* | RRID:AB_2567243 | (1:200) |
| Antibody | Anti-Fmi (mouse monoclonal) | DSHB #74 | RRID:AB_2619583 | (1:20) |
| Antibody | Anti-IR40a (guinea pig polyclonal) | *Silbering et al., 2011* | | (1:200) |
| Antibody | Anti-IR64a (rabbit polyclonal) | *Ai et al., 2010* | RRID:AB_2566854 | (1:1000) |
| Antibody | Anti-IR75a (rabbit polyclonal) | *Prieto-Godino et al., 2017* | RRID:AB_2631091 | (1:200) |

*Continued on next page*

*Continued*

| Reagent type (species) or resource | Designation | Source or reference | Identifiers | Additional information |
|---|---|---|---|---|
| Antibody | Anti-IR75b (guinea pig polyclonal) | *Prieto-Godino et al., 2017* | RRID:AB_2631093 | (1:500) |
| Antibody | Anti-IR75c (rabbit polyclonal) | *Prieto-Godino et al., 2017* | RRID:AB_2631094 | (1:200) |
| Antibody | Anti-Bruchpilot (mouse monoclonal) | DSHB nc82 | RRID:AB_2314866 | (1:10) |
| Antibody | Anti-Cadherin-N (rat monoclonal) | DSHB Ex#8–2 | RRID:AB_528121 | (1:25) |
| Antibody | Anti-Elav (rat monoclonal) | DSHB 7E8A10 | RRID:AB_528218 | (1:10) |
| Antibody | Anti-GFP (chicken polyclonal) | Abcam 13970 | | (1:2000) |
| Antibody | Anti-GFP (mouse monoclonal) | Invitrogen A11120 | | (1:1000) |
| Antibody | Anti-RFP (rabbit polyclonal) | Abcam 62341 | | (1:1000) |
| Antibody | Alexa488 (goat anti-guinea pig) | Invitrogen A11073 | | (1:500) |
| Antibody | Alexa488 (goat anti-chicken) | Abcam 150169 | | (1:1000) |
| Antibody | Cy3 (goat anti-rabbit) | Milan Analytica AG 111-165-144 | | (1:1000) |
| Antibody | Cy3 (goat anti-mouse) | Milan Analytica AG 115-165-166 | | (1:1000) |
| Antibody | Cy5 (donkey anti-rat) | Jackson ImmunoResearch 712-175-153 | | (1:250) |
| Antibody | Anti-DIG-POD | Roche Diagnostics 11 207 733 910 | | (1:300) |
| Software algorithm | Fiji | Fiji | RRID:SCR_002285 | |

## *Drosophila* culture

Flies were maintained on a standard wheat flour/yeast/fruit juice diet at 25°C in 12 hr light:12 hr dark conditions. Published mutant and transgenic *D. melanogaster* strains are described in the Key Resources Table. For the temporal control of *pdm3* RNAi with Gal80ts, animals of the desired genotype were cultivated continuously at 19°C or 29°C, or shifted from 19°C to 29°C after eclosion to permit induction of RNAi in adults; for all conditions, antennae or brains were dissection from 1-week-old flies. For the temporal control of *fmi* RNAi with Gal80ts, flies of the desired genotype were cultivated at 19°C; animals were staged by selecting white pupae (designated as 0 hr APF) and shifting these animals to 29°C after 10, 20, or 40 hr, to permit induction of RNAi at different time-points during pupal development. Unless otherwise indicated for specific experiments, antennae or brains were collected from both sexes.

## Transgenic flies

New transgenes were constructed using standard molecular biological procedures and transgenesis was performed by BestGene Inc using the phiC31 site-specific integration system. *Ir75a-CD4:tdGFP* (*Ir75a-GFP*) was generated by subcloning the 5′ (7833 bp) and 3′ (1921 bp) genomic sequences from the *Ir75a-Gal4* transgene (*Silbering et al., 2011*) to flank *CD4:tdGFP* in *pDESTHemmarG* (Addgene

31221) (*Han et al., 2011*), and integrated into the attP40 landing site. *Ir75c-CD4:tdGFP* (*Ir75c-GFP*) was generated by subcloning the 5' (3072 bp) and 3' (2545 bp) genomic sequences from the *Ir75c-Gal4* transgene (*Prieto-Godino et al., 2017*) to flank *CD4:tdGFP* in *pDESTHemmarG* and integrated into attP40. *Ir75b-CD4:tdTom* (*Ir75b-Tom*) was generated by cloning the 299 bp genomic sequence immediately upstream of the *Ir75b* start codon (as in the previously-described *Ir75b-GFP* transgene [*Prieto-Godino et al., 2017*]) into *pDESTHemmarR* (Addgene 31222) and integrated independently into attP5 and attP2 (on chromosome II and III, respectively). *Ir31a-CD4:tdTom* (*Ir31a-Tom*) was generated by cloning the 2000 bp genomic sequence immediately upstream of the *Ir31a* start codon, via Gateway recombination, into pDESTHemmarR, and integrated into attP2. *Ir84a-CD4:tdGFP* (*Ir84a-GFP*) was generated by cloning the 1964 bp genomic sequence immediately upstream of the *Ir84a* start codon, via Gateway recombination, into *pDESTHemmarG* and integrated into VK00027.

## TaDa sample preparation

Antennae from ~2000 flies (~1–8 days old, mixed sexes) of the desired genotype were harvested by snap-freezing animals in a mini-sieve (Scienceware, Bel-Art Products) with liquid nitrogen, shaking flies to detach and collect appendages in a Petri dish under the sieve containing Triton X-100 (0.1%). Third antennal segments were selected under a binocular microscope by pipetting and transferred to a 1.5 ml Eppendorf tube. After brief rinsing in PBS to remove detergent, antennae (occupying a volume of ~50–70 µl) were resuspended in 50 µl PBS and homogenized manually with a blue pestle (Sigma Aldrich Z359947-100EA) for at least 2 min. Antennal fragments were rinsed off the pestle with 100 µl PBS and 40 µl EDTA (100 mM) and 20 µl RNase (12.5 µg/µl) (Qiagen 19101) were added to the antennal lysate, mixed and allowed to sit for 3–5 min. 20 µl Proteinase K (Qiagen DNeasy Blood and Tissue Kit 69504) and 200 µl Buffer AL were added, and mixed by gently pipetting up and down with a blue tip ~50 times, before incubation in a 56°C heat block for 30 min. The digested lysate was centrifuged at 10,000 rpm for 30 s and the supernatant transferred to a new tube and allowed to cool before proceeding with DNA extraction.

Subsequent sample processing steps for TaDa – that is, *DpnI* treatment (which cuts at adenine-methylated GATC sites), DamID adaptor ligation, *DpnII* treatment (which cuts at non-methylated GATC sites, thereby digesting unmethylated DNA fragments), and PCR amplification and purification of methylated GATC fragments – were performed essentially as described (*Marshall et al., 2016*). PCR products were sonicated using a Covaris S220 to obtain ~300 bp average fragment size; DNA size and quality were verified by Qubit quantification. DamID adaptors were removed by *Sau3AI* digestion. Truseq Nano libraries were prepared at the Lausanne Genomic Technologies Facility and sequenced (SR 100 bp) on a HiSeq 2500.

## TaDa data analysis

To test for an enrichment of the genome occupancy of Dam:PolII relative to Dam alone, we used the 'damidseq pipeline' v1.4 (*Marshall and Brand, 2015*), with release 6 of the *D. melanogaster* genome as the reference. Within this pipeline, Bowtie v2.2.6 (*Langmead and Salzberg, 2012*) was implemented to align the Illumina sequence reads to the reference, and data processing called on SAMtools v1.2 (*Li et al., 2009*) and BEDTools v2 (*Quinlan and Hall, 2010*). This pipeline outputs log2 ratio files in bedgraph format. We calculated Pearson's Correlation Coefficient among triplicate datasets based on the bedgraph files, and also used them to generate a single averaged OSs file for each of the seven neuron population data sets. The files containing the averaged OSs were then inputted into the 'polii.gene.call' script (https://github.com/owenjm/polii.gene.call; version 1.0.2, *Marshall and Brand, 2015*; *Southall et al., 2013*) for estimating an FDR for each gene's OS. A file detailing the workflow and corresponding code is available on GitLab (https://gitlab.com/roman.arguello/ir_tada).

*Ir*, *Or*, *Gr*, and *Obp* genes analyzed are listed in *Supplementary file 4*. The set of genes encoding putative neuropeptides, GPCRs, ion channels (excluding *Ir*s, *Or*s, and *Gr*s), and neuron projection guidance proteins were based on annotated searches of Flybase (FB2020_06) using the following terms: 'neuropeptide', 'ion channel', 'GPCR', 'neuron projection guidance' (*Supplementary files 6*, *9*, *12*, *15*). The set of genes encoding transcription factors was based on a custom filtered set from the FlyTF.org database v2 (*Pfreundt et al., 2010*; *Supplementary file 18*).

For testing the hypothesis that OSs are equal among datasets (neuron populations), we used DESeq2 (*Love et al., 2014*). Unlike the DamID pipeline, this approach focused on individual GATC fragments within each gene. To quantify read coverage of each fragment, we used the same bam files that were outputted by the DamID pipeline (see above). We converted these to bed files using BEDTools v2 'bamToBed' utility and calculated the coverage of GATC fragments using the Bedtools' 'coverage' utility (*Quinlan and Hall, 2010*). The threshold for significant PolII occupancy within a dataset was a $\log_2$ fold-change of 1.5 and an adjusted *p*-value of 0.05. Differential occupancy was tested over the full dataset using a Likelihood-ratio test, where the full model specified neuron populations and DamID condition (Dam:PolII vs. Dam-alone) as factors, and an interaction term (between DamID condition and cell population): ~ cell.pop + cond + cond:cell.pop. The reduced model removed the interaction term (~ cell.pop + cond). This analysis yielded the 1694 genes analyzed in *Figure 4C–D*. We also carried out pairwise tests using a Wald test within DESeq2. Genes that were identified as candidates for differential expression were required to have ≥2 GATC fragments contributing to the difference (and to be significantly occupied by PolII).

Gene Ontology (GO) enrichment analyses were performed on the subset of genes that showed differential occupancy using clusterProfiler (v3.10.1) (*Yu et al., 2012*). We compared these genes against the three independent, controlled vocabularies provided by GO Consortium that model Biological Processes, Cellular Component, and Molecular Function (*Ashburner et al., 2000*).

## CATaDa analysis

CATaDa was performed on the seven populations of neurons using the same sets of sequencing reads (i.e. Dam-alone data) acquired for the TaDa analysis. The bioinformatics pipeline followed the general concepts of a previously described pipeline (*Aughey et al., 2018*), but with modifications. For each neuron population, the genomic sequencing reads from all three biological replicates were aligned to release 6.21 of the *D. melanogaster* genome with 'very-sensitive' settings (Bowtie2 v2.3.0). Following pooling of aligned reads from the replicates (samtools v1.7) (*Li et al., 2009*), the chromatin accessibilities were represented as the coverage of reads per bin of 10 bases, scaled to 1× average genomic coverage (~142.57 Mb) and averaged over a sliding window of 100 bases using bamCoverage within the deepTools library (v3.1.3) (*Ramírez et al., 2014*). The chromatin accessibility profile was plotted on genome browser tracks with pyGenomeTracks (v2.1) within the deepTools library. Significantly accessible regions were identified with peak-caller (MACS2, v2.1.1.20160309) (*Gaspar, 2018*) at an FDR cutoff (q-value) of 0.05. The correlation clustering of peaks among all assayed neuron populations was analyzed using the multiBamSummary program within the deep-Tools library (v3.1.3) with the Pearson method and bin size of 3000 bases.

## Histology

Immunofluorescence and RNA FISH on whole-mount antennae or antennal cryosections were performed as described (*Saina and Benton, 2013*). Immunofluorescence on whole-mount brains was performed essentially as described (*Ostrovsky et al., 2013*; *Silbering et al., 2011*). Antibodies used are listed in the Key Resources Table. RNA probes for *Ir31a* and *Ir84a* were previously described (*Benton et al., 2009*).

Quantification of En and Pdm3 immunofluorescence signals in OSN soma nuclei (*Figure 2—figure supplement 1B–C* and *Figure 5A–B*) was performed in Fiji (*Schindelin et al., 2012*). In brief, regions-of-interest (ROIs) were manually defined using the GFP signal (which circumscribes the nucleus) in a single optical slice where the cross-section of a given nucleus has the largest area, followed by automated measurement of the average En or Pdm3 immunofluorescence signal within this ROI. Background fluorescence signals in the tissue were measured in the same way within equivalently sized, arbitrarily chosen ROIs that did not overlap with GFP signals or any other OSN nuclei (visualized with DAPI).

## Statistical analysis

Preliminary experiments were used to assess variance and determine adequate sample sizes in advance of acquisition of the reported data; no statistical approaches were used to predetermine sample size. Details on specific statistical tests for individual experiments are provided in the corresponding figure legend. Unless stated otherwise, statistical significance thresholds were p<0.05 and

the tests performed were two-tailed. Measurements were taken from distinct samples, unless otherwise stated, along with a corresponding correction for multiple tests.

## Data availability

TaDa sequencing data and experimental information have been deposited in the ArrayExpress database at EMBL-EBI (https://www.ebi.ac.uk/arrayexpress/) and are available under accession number E-MTAB-8935. Processed data and intermediate files, including the read coverage at GATC motifs, are available at GitLab (https://gitlab.com/roman.arguello/ir_tada). Raw image data for *Figures 1* and *5–8*, and the figure supplements are available upon request.

## Code availability

A detailed workflow for the TaDa analyses and associated code and data files are available at GitLab (https://gitlab.com/roman.arguello/ir_tada; *Arguello, 2021*; copy archived at swh:1:rev: 2c181d8e15257d2344f4a0187ec3cf3be3e4b583).

## Biological material availability

All unique biological materials generated in this work are available from the corresponding author upon request.

## Acknowledgements

We thank Cheng-Ting Chien, Ya-Hui Chou, Fisun Hamaratoglu, Vladimir Katanaev, Liqun Luo, Tadashi Uemura, the Bloomington *Drosophila* Stock Center (NIH P40OD018537), the Vienna *Drosophila* Resource Center, and the Developmental Studies Hybridoma Bank (NICHD of the NIH, University of Iowa) for reagents. Sequencing was performed at the Lausanne Genomic Technologies Facility, and the Swiss Institute of Bioinformatics Vital-IT group provided computational resource support. We are grateful to Andrea Brand's lab for assistance in establishing the antennal TaDa protocol, Tony Southall and Owen Marshall for advice on the DamID analysis pipeline, Simon Anders for suggestions on DESeq2 usage, Thomas Hummel for discussions on OSN axon guidance, and members of the Benton laboratory for comments on the manuscript. J.R.A. was supported by a post-doctoral fellowship from the Novartis Foundation for medical-biological Research (12A14) and is currently supported by a Swiss National Science Foundation Assistant Professorship (PP00P3 176956). Research in R.B.'s laboratory was supported by the University of Lausanne, ERC Consolidator and Advanced Grants (615094 and 833548, respectively) and the Swiss National Science Foundation (310030B 185377)

## Additional information

### Funding

| Funder | Grant reference number | Author |
| --- | --- | --- |
| Novartis Foundation for Medical-Biological Research | 12A14 | J Roman Arguello |
| FP7 Ideas: European Research Council | 615094 | Richard Benton |
| H2020 European Research Council | 833548 | Richard Benton |
| Schweizerischer Nationalfonds zur Förderung der Wissenschaftlichen Forschung | 310030B 185377 | Richard Benton |
| Swiss National Science Foundation | PP00P3 176956 | J Roman Arguello |

The funders had no role in study design, data collection and interpretation, or the decision to submit the work for publication.

## Author contributions
J Roman Arguello, Conceptualization, Resources, Data curation, Software, Funding acquisition, Validation, Investigation, Visualization, Methodology, Project administration, Writing - review and editing, Contributed Figures 1D–G, 2A–G, 4B–E, 5B, 7C, 8E, Figure 1—figure supplement 1–4, Figure 2—figure supplement 1A,C; Liliane Abuin, Validation, Investigation, Visualization, Writing - review and editing, Contributed Figures 1B–C, 5A–C, 7C, 8A–E, Figure 2—figure supplement 1B–C, Figure 7—figure supplement 1B–C, Figure 8—figure supplement 1; Jan Armida, Resources, Validation, Investigation, Visualization, Writing - review and editing, Contributed Figures 7A–B, 8A–C, 8E–F, Figure 7—figure supplement 1D; Kaan Mika, Validation, Investigation, Visualization, Writing - review and editing, Contributed Figures 5C–D, 6, Figure 7—figure supplement 1A; Phing Chian Chai, Resources, Software, Validation, Investigation, Visualization, Writing - review and editing, Contributed Figures 3, 4C; Richard Benton, Conceptualization, Supervision, Funding acquisition, Visualization, Writing - original draft, Project administration, Writing - review and editing

## Author ORCIDs
J Roman Arguello ⓘ https://orcid.org/0000-0001-7353-6023
Richard Benton ⓘ https://orcid.org/0000-0003-4305-8301

## Decision letter and Author response
Decision letter https://doi.org/10.7554/eLife.63036.sa1
Author response https://doi.org/10.7554/eLife.63036.sa2

# Additional files

## Supplementary files
• Supplementary file 1. Full set of genes with significant OSs in Ir neuron populations. OSs correspond to those summarized in *Figure 2A*.

• Supplementary file 2. List of gene IDs only from *Supplementary file 1*.

• Supplementary file 3. Genes with significant OSs unique to each Ir neuron population. IDs correspond to the counts displayed in *Figure 2A*.

• Supplementary file 4. Set of chemosensory receptor and *Obp* genes.

• Supplementary file 5. Chemosensory receptor and *Obp* genes with significant OSs in Ir neuron populations. Only significant OSs are listed. Values correspond to those plotted in *Figure 1—figure supplements 1–4*. NA = not applicable (i.e. where there is non-significant occupancy for a given gene in a particular neuron population).

• Supplementary file 6. Set of neuropeptide genes. Genes correspond to those in *Figure 2C*.

• Supplementary file 7. Neuropeptide genes with significant OSs in Ir neuron populations. IDs correspond to the counts displayed in *Figure 2C*.

• Supplementary file 8. Neuropeptide genes with significant OSs unique to each Ir neuron population. IDs correspond to the counts displayed in *Figure 2C*.

• Supplementary file 9. Set of G protein-coupled receptor genes. Genes correspond to those in *Figure 2D*.

• Supplementary file 10. G protein-coupled receptor genes with significant OSs in Ir neuron populations. IDs correspond to the counts displayed in *Figure 2D*.

• Supplementary file 11. G protein-coupled receptor genes with significant OSs unique to each Ir neuron population. IDs correspond to the counts displayed in *Figure 2D*.

• Supplementary file 12. Set of ion channel genes. Genes correspond to those in *Figure 2E*.

• Supplementary file 13. Ion channel genes with significant OSs in Ir neuron populations. IDs correspond to the counts displayed in *Figure 2E*.

• Supplementary file 14. Ion channel genes with significant OSs unique to each Ir neuron population. IDs correspond to the counts displayed in *Figure 2E*.

• Supplementary file 15. Set of neuron projection guidance molecule genes. Genes correspond to those in *Figure 2F*. Note this list encompasses genes encoding molecules potentially directly or indirectly involved in neural guidance, and includes some neuronally-expressed transcription factors (such as *pdm3*, the unique marker of Ir75a neurons).

• Supplementary file 16. Neuron projection guidance molecule genes with significant OSs in Ir neuron populations. IDs correspond to the counts displayed in *Figure 2F*.

• Supplementary file 17. Neuron projection guidance molecule genes with significant OSs unique to each Ir neuron population. IDs correspond to the counts displayed in *Figure 2F*.

• Supplementary file 18. Filtered set of transcription factor genes. Genes correspond to those in *Figure 2G*.

• Supplementary file 19. Transcription factor genes with significant OSs in Ir neuron populations. IDs correspond to the counts displayed in *Figure 2G*.

• Supplementary file 20. Transcription factor genes with significant OSs unique to each Ir neuron population. IDs correspond to the counts displayed in *Figure 2G*.

• Transparent reporting form

## Data availability

All data generated or analyzed during this study are included in the manuscript and supporting files, or in data repositories as specified in the corresponding methods section.

The following dataset was generated:

| Author(s) | Year | Dataset title | Dataset URL | Database and Identifier |
|---|---|---|---|---|
| Arguello JR, Abuin L, Armida J, Mika K, Chai PC, Benton R | 2021 | Sequencing of methylated GATC DNA fragments (Targeted DamID) from seven olfactory neuron populations in *Drosophila melanogaster* | https://www.ebi.ac.uk/arrayexpress/experiments/E-MTAB-8935/ | ArrayExpress, E-MTAB-8935 |

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
