## [Decision Letter]

**Acceptance summary:**

The work describes a novel and robust method to determine differentially expressed genes in specific classes of *Drosophila* olfactory sensory neurons (OSNs) using the Targeted DamID technique. This is valuable as it provides an approach that can be used to profile transcriptions in cell types in the antenna that are difficult to isolate and purify. Through such an analysis and additional experiments to determine the function of some OSN-specific mRNAs, this study not only provides an important resource for the OSN field (especially for *Drosophila* and mosquito researchers) but also provides mechanistic insight into the determination of specific OSN classes, and how they are segregated into specific glomeruli.

**Decision letter after peer review:**

Thank you for submitting your article "Targeted molecular profiling of rare cell populations identifies olfactory sensory neuron fate and wiring determinants" for consideration by *eLife*. Your article has been reviewed by three peer reviewers, and the evaluation has been overseen by a Reviewing Editor and Utpal Banerjee as the Senior Editor. The following individual involved in review of your submission has agreed to reveal their identity: Tony D Southall (Reviewer #1).

The reviewers have discussed the reviews with one another and the Reviewing Editor has drafted this decision to help you prepare a revised submission.

Summary:

Arguello et al. report comprehensive transcriptional (and chromatin accessibility states) profiles for individual classes of *Drosophila* olfactory sensory neurons (OSNs), uncovered using the Targeted DamID technology. Seven individual IR-expressing OSN classes (Ir64a, Ir31a, Ir75a, Ir75b, Ir75c and Ir84a) were profiled by using IR-specific GAL4 drivers to control the expression of Dam (CATaDa) or Dam:Pol II (TaDa) which indicate chromatin accessibility and RNA polymerase II occupancy respectively. The data correlates well with what is already known about these OSNs and complement two other studies in Current biology and in BioRxiv, which use single-cell RNAseq to profile mRNA expression in individual OSNs. They describe differentially expressed transcription factors and cell surface molecules and follow up nicely with a functional analysis of the transcription factor *pdm3*, which is expressed specifically in Ir75a OSNs and the cell adhesion molecule fmi, expressed in most Ir expressing OSNs except for Ir75a and Ir84a. These studies show developmental functions consistent with their differential expression (documented in adults): most interesting that mutations in fmi are associated with disruption of glomerular organization for all coeloconic OSNs.

Overall, the study is of high technical quality and will be appreciated by a broad audience for its technical breakthrough and available datasets. Not only does this study provide an important resource for the OSN field (especially for *Drosophila* and mosquito researchers) but also provides mechanistic insight into the determination of specific OSN classes, and how they are segregated into specific glomeruli. Also, the successful application of Targeted DamID, in this context, emphasises its effectiveness for profiling small and difficult to access populations of cells in vivo.

There are however, several points that should addressed before publication.

Essential revisions:

Readers would benefit from detailed comparisons of the data presented here with adult OSN single cell profiling data from the Luo lab (McLaughlin et al., 2020) and a previous study published in Current Biology early 2020. While carefully comparing and contrasting these results with the Luo preprint, the authors should acknowledge and discuss inconsistencies such as expression of odorant binding proteins, which are presented as originating from support cells of olfactory sensilla (in this study) versus some being expressed by OSNs (McLaughlin et al. study). Also, the Discussion needs to be modified as the scRNAseq seems to be feasible for adult OSNs (McLaughlin et al. study).

The focus on Pdm3 and Fmi in the last part of the manuscript (Figures 5-7) does address the functional significance of their continued, heterogeneous expression in adult OSNs. Attempts to knockdown the expression of these genes in adult OSNs failed to yield significant phenotypes. In fact, the observed cell fate or wiring phenotypes require knocking down the genes in all OSNs at much earlier developmental stages (using *peb-GAL4*) than the time points when the datasets were generated. The Results and Discussion sections should be revised to clearly acknowledge that these experiments, though well designed and executed, do not connect to the TaDa datasets which pertain to adult expression (see also points 9 and 10). That said, the study is strong and interesting enough even with the first five figures.

The authors may consider revising the title to eliminate the focus on neuron fate and wiring given that the TaDa datasets are generated from adult OSNs but there is no obvious phenotype when knocking down Pdm3 and Fmi in adult OSNs.

Figure 1D and E: The DamID plots in Figure 1D and E look strange – why are there sloped boxes representing the data? Is this due to it representing the sequencing reads rather that the binned average per GATC fragment? Ideally, DamID data should only be displayed as the binned signal per GATC fragment (the regions between GATC sites), as this is the maximum resolution of the system.

It is not clear from the figure legend (Figure 1D and E) whether the read depth is normalised, and if so, how it was normalised.

The approximate positions of the GATC sites (would be good if they were exact) do not appear to always align with the distinct boundaries within the data, which would be at GATC sites.

Are there signature genes for each OSN class profiled other than the IRs themselves? Can these genes be statistically identified and provided as a list?

Figure 1—figure supplement 1 – it is very difficult to read the receptor names. Can the panels be rearranged (maybe by making the columns narrower), so that the receptor names are larger? How do the authors reconcile expression of ORs and GRs to be expressed in adult OSNs? One explanation by the authors is that these sensory receptor genes reside within introns or genomic regions that also have other genes known to express in neurons. Can the authors do an in situ hybridization to confirm that indeed some OR/GR genes are co-expressed with IRs?

Looking at the fmi mutant phenotype it seems like mostly the effect starts at 24 hours with defasciculated axons around future antennal lobe. A discussion of this should be added to the Discussion. Can the authors provide more discussion on how they envision Fmi working to regulate OSN projections in the antennal lobes?

It also seems like many of the OSNs analyzed in the coeloconic sensilla seem to have a defect. How do the authors reconcile this with the differential expression of fmi in IR expressing OSNs? It seems like the wiring of Ir84a and Ir75a OSNs also look affected even though they do not express fmi? is this non-autonomous? We note that reverse MARCM using regular fmi alleles could answer this. Or if RNAi needs to be used maybe a gal4 that is expressed early in Ir84a OSNs? In any event, this should be acknowledged as an issue remaining to be resolved.

Subsection “Targeted DamID of OSN populations”, the finding of low RNA PolymeraseII occupancy upstream of Ir93a versus high protein expression: Maybe a mention of TaDa not as effective for profiling low abundance gene expression?

In Figure 6E, it seems like early and wide spread knock down of *pdm3* results in Ir75a OSNs to be converted to Ir75b OSNs, which leads to an increased glomerular volume targeted by Ir75b OSNs. What happens to projection patterns in the experiment presented in 6G where RNAi knock down of *pdm3* is driven by *Ir8aGAL4*, which leads to OSNs that co-express Ir75a and Ir75b?

The representation of the CATaDa data in Figure 3 gives a false impression of the resolution of the data – it should be binned per GATC fragment.

Figure 5B – it would be good to see some statistical tests for these data.

Materials and methods (“TaDa sample preparation”): Should indicate the sex of the experimental flies.

---

## [Author Response]

Essential revisions:Readers would benefit from detailed comparisons of the data presented here with adult OSN single cell profiling data from the Luo lab (McLaughlin et al., 2020) and a previous study published in Current Biology early 2020. While carefully comparing and contrasting these results with the Luo preprint, the authors should acknowledge and discuss inconsistencies such as expression of odorant binding proteins, which are presented as originating from support cells of olfactory sensilla (in this study) versus some being expressed by OSNs (McLaughlin et al. study). Also, the Discussion needs to be modified as the scRNAseq seems to be feasible for adult OSNs (McLaughlin et al. study).

We fully agree with the interest to compare TaDa and single cell/nucleus-RNA-seq data. The most relevant study is the adult OSN single-nucleus RNA-seq study from McLaughlin et al., but – as discussed already with the editor – the datasets for this preprint are not in the public domain at the time of revision, which limits our ability to extract detailed information from this work. We also note that the neuronal populations characterized by RNA-seq and by TaDa have only limited overlap, constraining direct comparison of cell types. While comparative assessment of TaDa and sc/nRNA-seq data will likely be most fruitful from equivalent classes of cells and developmental timepoints, in our revised manuscript we have expanded our Discussion comparing and contrasting general properties of the OSN gene expression profiles emerging from these different approaches. We further note that the snRNA-seq datasets led to clustering of neurons expressing *Ir75a*, *Ir75b* and *Ir75c* (as well as Or35a neurons), which highlights one advantage of our targeted approach in selectively distinguishing these cell populations. This was of particular importance in our study, as it led us to discover the specific expression and function of Pdm3 in Ir75a neurons. (Of course, future targeted sc/nRNA-seq of individual OSN classes is certainly conceivable).

Regarding the intriguing finding of OBP expression by McLaughlin et al., we have now analyzed the TaDa OSs for the entire OBP repertoire (new Figure 1—figure supplement 4), but we do not detect significant occupancy of these genes except in rare cases. For example, *Obp19d* (pan-neuronally expressed in McLaughlin et al.) is only occupied in Ir75b neurons in our TaDa datasets, and *Obp56d* (expressed in most neurons in McLaughlin et al.) is not occupied in any neuron. Our TaDa data are consistent with previous in situ RNA and protein expression analyses of OBPs (in *Drosophila* and many other insect species) which have reported expression in support cells but not in neurons (e.g., PMID 27845621). It is certainly possible that snRNA-seq is much more sensitive than either TaDa or in situ analysis of *Obp* expression. However, as extensive genetic analysis of *Obp*s has revealed little or no physiological phenotypic consequences (PMID 31651397), the functional significance of neuronal *Obp* expression remains unclear. We discuss these issues in the revised manuscript.

Finally, in Figure 2, we have also improved the presentation of OSs of sets of known/predicted ion channels, neuropeptides, GPCRs and axon guidance molecules, which were previously lumped together in the “cell surface and secreted protein” (CSSP) gene set. This aligns our TaDa data presentation somewhat better with the McLaughlin et al. analysis of channels and neuropeptide/neurotransmitter signaling systems (Figure 9 of that preprint). As most of these genes are broadly expressed across neuron populations, and because of the limited number of neuron types that we can directly compare between TaDa and RNA-seq data, we felt that in-depth comparison would not yield meaningful conclusions at this stage. However, we hope our more refined functional classifications will help in the future exploration of these data.

The focus on Pdm3 and Fmi in the last part of the manuscript (Figures 5-7) does address the functional significance of their continued, heterogeneous expression in adult OSNs. Attempts to knockdown the expression of these genes in adult OSNs failed to yield significant phenotypes. In fact, the observed cell fate or wiring phenotypes require knocking down the genes in all OSNs at much earlier developmental stages (using peb-GAL4) than the time points when the datasets were generated. The Results and Discussion sections should be revised to clearly acknowledge that these experiments, though well designed and executed, do not connect to the TaDa datasets which pertain to adult expression (see also points 9 and 10). That said, the study is strong and interesting enough even with the first five figures.

We acknowledge that the TaDa experimental design in this study captures gene expression profiles of OSNs in late developmental stages, but stress that although we harvested adult antennae, it is likely that the Dam methylation patterns reflect, at least in part, Dam:PolII occupancy from the mid-pupal stage when expression of this fusion protein is induced, rather than that of adult OSNs exclusively. (To our knowledge, it is unclear if Dam-dependent methylation is a permanent mark of occupancy or if it is turned over). We suspect that significant TaDa OSs represent both genes that are expressed and function primarily in mature OSNs (such as the *Ir*s) as well as those that act during development. Fmi is one such example, as the adult expression (detected by immunofluorescence) is very low (Figure 7A). Our more refined analysis of OSs of different gene classes illustrates that many other known or putative neuronal guidance molecules can be detected by TaDa (Figure 2F). These observations presumably reflect the persistent expression of such molecules, albeit at lower levels, in late developmental stages, and is consistent with previous bulk RNA-seq datasets of adult antennae (e.g., PMID 25412082, 30102700).

The transcription factor we functionally characterized in our work, Pdm3, has persistent protein expression in Ir75a neurons in adult stages (Figure 5A-B). Moreover, our new temporally-controlled RNAi experiments (described in detail below) strengthen our original claim that Pdm3 is continuously required in adult antennae to suppress Ir75b receptor expression in Ir75a neurons, in addition to its earlier role in distinguishing the fate of Ir75a and Ir75b neurons.

The authors may consider revising the title to eliminate the focus on neuron fate and wiring given that the TaDa datasets are generated from adult OSNs but there is no obvious phenotype when knocking down Pdm3 and Fmi in adult OSNs.

As described in the comment above, we believe our TaDa datasets are relevant for identifying genes with roles during development and/or in adult stages. To effectively encompass the scope of the analyses and functional follow-up in our work, we have revised the title to “Targeted molecular profiling of rare olfactory sensory neurons identifies fate, wiring and functional determinants”.

Figure 1D and E: The DamID plots in Figure 1D and E look strange – why are there sloped boxes representing the data? Is this due to it representing the sequencing reads rather that the binned average per GATC fragment? Ideally, DamID data should only be displayed as the binned signal per GATC fragment (the regions between GATC sites), as this is the maximum resolution of the system.

The reviewers are correct that this panel shows read depth and not binned averaged depth at GATC fragments. The presentation format in this panel was chosen to give a completely “raw” look of the dataset prior to processing.

It is not clear from the figure legend (Figure 1D and E) whether the read depth is normalised, and if so, how it was normalised.

As described above, the data is raw and not normalized. We have added in Figure 1E a graphical element to clarify the workflow.

The approximate positions of the GATC sites (would be good if they were exact) do not appear to always align with the distinct boundaries within the data, which would be at GATC sites.

We have now added the exaction location of the GATC positions, which align well with the distinct boundaries.

Are there signature genes for each OSN class profiled other than the IRs themselves? Can these genes be statistically identified and provided as a list?

There are indeed signature genes for the neuron classes profiled; for example, amongst the transcription factor genes *pdm3* was the unique marker for Ir75a neurons, which is why we focused on it for functional analysis. We have now expanded our analysis and documentation of signature genes for various categories of molecules (ion channels, GPCRs, neuropeptides, neural guidance molecules, in addition to transcription factors and a genome-wide list). These data are summarized in Figure 2 and full lists are provided in the accompanying supplementary files. For particular functional categories there are generally (but not always) very few unique genes for a given neuron population; these will be interesting candidates for future in situ validation of expression and functional studies.

Figure 1—figure supplement 1 – it is very difficult to read the receptor names. Can the panels be rearranged (maybe by making the columns narrower), so that the receptor names are larger? How do the authors reconcile expression of ORs and GRs to be expressed in adult OSNs? One explanation by the authors is that these sensory receptor genes reside within introns or genomic regions that also have other genes known to express in neurons. Can the authors do an in situ hybridization to confirm that indeed some OR/GR genes are co-expressed with IRs?

To improve readability, we have now split this figure into three for each of the main receptor families, as well as adding an additional equivalent figure supplement for the *Obp* genes, as mentioned above. Furthermore, the color shading of these heatmaps have been unified with those in the main figures.

Regarding the *Or* and *Gr* genes displaying significant occupancy scores in certain Ir neuron populations (Figure 1—figure supplements 2-3), numerous previous studies (e.g., PMID 10943836, 16139208, 16139209, 11257221) have comprehensively surveyed expression of these genes using RNA (F)ISH and transgenic promoter reporter approaches. These analyses do not provide evidence for the expression of the occupied *Or*s or *Gr*s in the spatial domains of the antenna characteristic of Ir neurons and/or in neurons that target known “Ir glomeruli” in the antennal lobe. For example, *Gr61a* and *Gr63a* have significant occupancy in 4 or 5 Ir neuron populations, but their endogenous expression is clearly limited to the basiconic OSN population projecting to the V glomerulus (PMID 17167414, 17360684). Bulk RNA-seq analysis (PMID 25412082) or the snRNA-seq dataset of McLaughlin et al., also fail to offer evidence of *Or* or *Gr* transcripts in adult Ir neurons. Finally, we note that the obligate OR co-receptor gene *Orco* is not occupied in many of the populations in which other “tuning” *Or* genes have significant OSs, arguing that this occupancy is not likely to be functionally meaningful.

While we cannot rule out that there may be isolated examples of *Or* and *Ir* co-expression, we suggest that in the vast majority of cases, the significant OSs for *Gr*s and *Or*s is due to the inadequate resolution of TaDa to distinguish overlapping genes, as well as those that are tightly clustered in the genome (many sensory receptors are separated by only a few hundred bp from flanking non-receptor genes). We have added a comment on this latter point to the text.

Looking at the fmi mutant phenotype it seems like mostly the effect starts at 24 hours with defasciculated axons around future antennal lobe. A discussion of this should be added to the Discussion. Can the authors provide more discussion on how they envision Fmi working to regulate OSN projections in the antennal lobes?

We agree with the reviewers that the requirement for Fmi is likely to start early in development as, or even before, OSN axons start to invade the antennal lobe. While we do not want to speculate too far beyond the available data, we have added some additional discussion of the potential mechanism of action.

It also seems like many of the OSNs analyzed in the coeloconic sensilla seem to have a defect. How do the authors reconcile this with the differential expression of fmi in IR expressing OSNs? It seems like the wiring of Ir84a and Ir75a OSNs also look affected even though they do not express fmi? is this non-autonomous? We note that reverse MARCM using regular fmi alleles could answer this. Or if RNAi needs to be used maybe a gal4 that is expressed early in Ir84a OSNs? In any event, this should be acknowledged as an issue remaining to be resolved.

These are all excellent points. Based upon its broad expression across the antennal lobe, we hypothesize that Fmi has a widespread role in these neurons, and that our pan-OSN RNAi has both direct effects on neurons that express higher levels of this protein, and indirect (non-autonomous) roles on those with lower or absent Fmi expression. In preliminary (standard) MARCM experiments, removing Fmi from subsets of neurons (covering several different OSN populations) did not lead to striking glomerular phenotypes, which suggests that Fmi must be removed from all neurons of a given population in order to observe the neuronal segregation phenotype observed with the pan-RNAi, perhaps because of compensation by other cell surface molecules. As such, reverse MARCM, for which we do not have the necessary lines, is unlikely to be more informative at this stage. We would also very much like to have driver lines for population-specific early RNAi, but such tools are not yet available. As we do not have publication-quality data to add to the manuscript in the short-term, we prefer to acknowledge these outstanding important issues and potential future solutions in the expanded discussion of the role of Fmi in OSN axon guidance.

Subsection “Targeted DamID of OSN populations”, the finding of low RNA PolymeraseII occupancy upstream of Ir93a versus high protein expression: Maybe a mention of TaDa not as effective for profiling low abundance gene expression?

This comment prompted us to double-check the OS for *Ir93a*; this is in fact significant (albeit low) in Ir40a neurons, but there is also a significant OS in four other Ir neuron populations which do not express IR93 protein (as detected by our antibody) (Supplementary file 5). We suspect that bona fide transcription of *Ir93a* in Ir40a neurons is too low to be detected by TaDa, and that the significant OSs we detect across these five populations is artefactual, potentially reflecting the expression of a ∼5kb *Doc* transposable element (PMID: PMID: 7813908) in an intron.

We add that we do not know if Ir93a protein is highly expressed; we say it is “readily detected” but this could be because the antibody has extremely high-affinity. At present we cannot make any confident statement about the limits of sensitivity of TaDa, as this would require parallel profiling of the same tissue/developmental stage by TaDa and RNA-seq.

We have amended our discussion of Ir93a in the Results.

In Figure 6E, it seems like early and wide spread knock down of pdm3 results in Ir75a OSNs to be converted to Ir75b OSNs, which leads to an increased glomerular volume targeted by Ir75b OSNs. What happens to projection patterns in the experiment presented in 6G where RNAi knock down of pdm3 is driven by Ir8aGAL4, which leads to OSNs that co-express Ir75a and Ir75b?

We have performed several additional *pdm3* RNAi experiments to address the temporal requirement of this transcription factor in Ir75a neurons.

We previously used *Ir8a-Gal4* to drive late pupal/adult RNAi, which led to a relatively mild phenotype (i.e., a slight increase in the number of OSNs expressing IR75b, no change in Ir75a neuron number) compared to the results with *peb-Gal4*. In new experiments, we temporally-restricted RNAi exclusively to the adult stage by combining *peb-Gal4* with the temperature-sensitive inhibitor *Gal80^ts^*. Importantly, adult-only *peb>pdm3^RNAi^* animals exhibit increases and decreases in the number of neurons expressing Ir75b or Ir75a, respectively, that are similar to our observations with *peb>pdm3^RNAi^* throughout development (new Figure 6F, compared with Figure 5D). We also detect many neurons co-expressing IR75b and IR75a (Figure 6G-H), as observed (less frequently) in the original *Ir8a>pdm3^RNAi^* experiments. The ectopic expression of IR75b in Ir75a neurons in adult-only *pdm3^RNAi^* was confirmed by visualizing *Ir75b-GFP*: this reporter labels both the normal DL2d glomerulus and, ectopically, the DP1l (Ir75a) glomerulus (Figure 6I), consistent with misexpression of *Ir75b-GFP* in Ir75a neurons *after* they have targeted to DP1l. This phenotype contrasts with that of pan-developmental *peb>pdm3^RNAi^* (Figure 6E) where loss of *pdm3* also leads to a change in glomerular targeting of “Ir75a” neurons. These observations confirm and strengthen our original model in which Pdm3 has an early role in fate determination of Ir75a neurons (by preventing them from adopting the receptor expression profile and glomerular wiring properties of Ir75b neurons), and an ongoing role in adults to repress Ir75b expression.

In response to the reviewers’ specific question, we have also combined *Ir8a>pdm3^RNAi^* with the *Ir75a-GFP* and *Ir75b-GFP* reporters. *Ir75a-GFP*-labelled axons can still be detected innervating the normal glomerular target (DP1l) (Figure 6—figure supplement 1D), consistent with the lack of effect of this late knock-down of *pdm3* on endogenous Ir75a expression (Figure 6—figure supplement 1A). *Ir75b-GFP*-labelled axons display innervation only to DL2d. Although we had anticipated detecting some innervation of these neurons in DP1l (as in the experiments described above), there are two potential issues: (i) the *Ir8a>pdm3^RNAi^* phenotype (Figure 6—figure supplement 1A) is much weaker compared to *peb>pdm3^RNAi^*, and (ii) ectopic *Ir75b-GFP* transgene expression in antennae is already milder than ectopic IR75b protein expression, even in *peb>pdm3^RNAi^* (Figure 6D). We suspect that any ectopic *Ir75b*-*GFP* reporter expression in Ir75a neurons in *Ir8a>pdm3^RNAi^* animals is simply too sparse and/or weak to give a detectable signal in the axon termini some distance away in the antennal lobe.

Given that we have achieved stronger and more temporally-restricted adult-only *pdm3^RNAi^* with *peb-Gal4* and *Gal80^ts^* (albeit across most/all OSNs), we have now moved all of the phenotypic data with *Ir8a>pdm3^RNAi^* to a figure supplement, and we mention the issue of the weaker RNAi phenotype in the accompanying figure legend.

The representation of the CATaDa data in Figure 3 gives a false impression of the resolution of the data – it should be binned per GATC fragment.

For the CATaDa analysis we originally used the published script (from PMID 29481322) but found that this produced severely reduced signals at all of the specific genes of interest for our study. For this reason, we implemented the pipeline as described in the Materials and methods. We did consider binning the bam files via a GATC coverage file, but smoothing and duplicate read removal are inevitable because at any occupied site, the cut GATC at the boundaries can only be amplified from one direction (the “sister” fragments going towards the opposite direction are too long to be PCR amplified in most cases or too short to be retained). This leads to an uneven build-up of reads at both GATC ends. In our representation, each bump is flanked by two GATC sites. We stress that we are not claiming higher resolution, but it is expected that the region between two cut GATCs in close proximity will have higher probability of occupancy than the areas beyond the GATCs, which would explain the tapering of the signal. Binning the data based on GATC fragments (which would require us to devise a new pipeline) would result in a sharp fall of signal at the boundaries and this, too, would not be a true representation of occupancy, nor would it reflect the raw read data. We now emphasize this issue in the figure legend and stress the limits of resolution of the CATaDa data. We hope the reviewers will find it acceptable to maintain the same analysis method in this study.

Figure 5B – it would be good to see some statistical tests for these data.

We have now added the results of statistical test to these data (as well as for the equivalent experiments in Figure 2—figure supplement 1C).

Materials and methods (TaDa sample preparation): Should indicate the sex of the experimental flies.

We indicate in the “*Drosophila* culture” section that mixed sexes were used in all experiments (unless indicated otherwise), and we now re-iterate this in the “TaDa sample preparation” section.